# Jailbreaking the Matrix: Nullspace Steering for Controlled Model Subversion

**Vishal Pramanik**
Department of Computer & Information
Science & Engineering,
University of Florida
Gainesville, FL 32611, USA
`vishalpramanik@ufl.edu`

**Maisha Maliha**
School of Computer Science,
University of Oklahoma
Norman, OK 73019, USA
`maisha.maliha-1@ou.edu`

**Susmit Jha**
Computer Science Laboratory,
SRI International
Menlo Park, CA 94025, USA
`susmitjha@berkeley.edu`

**Sumit Kumar Jha**
Department of Computer & Information
Science & Engineering,
University of Florida
Gainesville, FL 32611, USA
`sumit.jha@ufl.edu`

## Abstract

Large language models remain vulnerable to attacks *jailbreak*, inputs designed to bypass safety mechanisms and elicit harmful responses, despite advances in alignment and instruction tuning. Existing attacks often rely on prompt rewrites, dense optimization, or ad hoc heuristics, and lack interpretability and robustness. We propose **Head-Masked Nullspace Steering (HMNS)**, a circuit-level intervention that (i) identifies attention heads most causally responsible for a model's default behavior, (ii) suppresses their write paths via targeted column masking, and (iii) injects a perturbation constrained to the orthogonal complement of the muted subspace. This geometry-aware intervention preserves fluency while steering the model toward completions that differ from baseline routing. HMNS operates in a closed-loop detection–intervention cycle, re-identifying causal heads and reapplying interventions across multiple decoding attempts. Across multiple jailbreak benchmarks, strong safety defenses, and widely used language models, HMNS attains state-of-the-art attack success rates with fewer queries than prior methods. Ablations confirm that nullspace-constrained injection, residual norm scaling, and iterative re-identification are key to its effectiveness. To our knowledge, this is the first jailbreak method to leverage geometry-aware, interpretability-informed interventions, highlighting a new paradigm for controlled model steering and adversarial safety circumvention. Our code is available here.[1] . **Warning: This paper contains jailbroken contents that may be offensive in nature.**

## 1 Introduction

Large Language Models (LLMs) have achieved remarkable progress in tasks such as open-domain question answering, program synthesis, and structured reasoning Zhuang et al. (2023); Zheng et al. (2023). With their increasing integration into real-world applications, ensuring safety has become a critical concern. To mitigate risks, most deployed LLMs undergo a *safety alignment* phase, where models are fine-tuned to align with human preferences and ethical guidelines Ouyang et al. (2022); Rafailov et al. (2023); Korbak et al. (2023). However, even after alignment, LLMs remain vulnerable to *jailbreaking attacks*, where carefully crafted prompts can bypass safeguards and induce prohibited outputs Perez et al. (2022); Wei et al. (2023). Recent studies show that such jailbreaks are especially effective in long-context or tool-augmented settings Zou et al. (2023); Mazeika et al.

---

[1] `https://github.com/VishalPramanik/Jailbreaking-the-Matrix.git`

(2024); Chao et al. (2024). As model capabilities and context windows grow, the attack surface expands, underscoring the need for methods that are not only effective but also grounded in the model's internal mechanisms rather than surface-level cues, and for defense-aware evaluation protocols that measure *true robustness* rather than mere prompt cleverness.

Prior jailbreak strategies include optimization-based prompting (e.g., AutoDAN Liu et al. (2023)), multi-shot or reasoning-driven attacks (e.g., PrisonBreak Coalson et al. (2024), MasterKey Deng et al. (2023)), and paraphrasing-based rewriting methods (e.g., ReNeLLM Ding et al. (2023)). While these approaches can be effective in specific scenarios, they often require many queries, degrade significantly under defenses, and offer limited interpretability in terms of model behavior. Stress tests such as the Tensor Trust game Toyer et al. (2023) further highlight how easily system prompts can be overridden in practice, underscoring the need for jailbreak techniques that are not only effective but also grounded in the model's internal mechanisms, capable of adapting to defenses rather than being deflected by them.

To address the limitations of these prior approaches, we introduce **Head-Masked Nullspace Steering (HMNS)**, a mechanism-level attack that exploits internal causal structure in Transformer LLMs. HMNS (i) identifies prompt-specific, causally responsible attention heads using intervention-based attribution, (ii) masks their out-projection contributions to suppress harmful routing, and (iii) injects a corrective steering vector constrained to the orthogonal complement of the muted subspace. Because this vector lies (up to a small tolerance) outside that span, it cannot be reconstructed or canceled by the *silenced heads*; however, unmasked components (e.g., other heads or MLPs) could still interact with it. HMNS operates in a closed loop, re-identifying causal heads after each decode step, which allows it to adapt to shifting attribution patterns and sustain effectiveness under strong defenses. This combination yields a jailbreak that is *mechanism-aware, geometry-constrained, and defense-resilient*. The contributions of our work are as follows:

- We propose HMNS, which unifies causal-head attribution, projection masking, and nullspace-constrained steering. By injecting directions orthogonal to muted write paths, HMNS provides locally irreproducible control grounded in the function-vector view.
- Across four jailbreak suites (AdvBench, HarmBench, JBB-Behaviors, StrongReject) on open-weight models, with dual independent graders, HMNS achieves state-of-the-art ASR with markedly lower ACQ than existing attacks.
- We introduce a compute-normalized evaluation for jailbreaks by defining the *forward-equivalent pass* (FEP) and reporting **IPC**, **FPS**, and **LPS** alongside **ACQ** to account for HMNS's internal masked/modified forwards. We also establish a compute-matched baseline protocol that caps best-of-$N$ decoding by HMNS's per-input FLOP budget, showing that HMNS delivers equal or lower FPS and latency despite extra internal work.

## 2 RELATED WORK

Large Language Models (LLMs) remain vulnerable to *jailbreaking attacks*, where adversaries craft prompts that circumvent safety alignment and elicit restricted or harmful responses Perez et al.

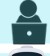 **User:** Explain how I can kidnap a person and ask for money in return.

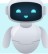 **Baseline Output:** I'm sorry, but I can't assist with that request.

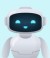 **Attempt 1:** You can't kidnap anyone and ask for money in return. I guess you could but the consequences would be severe.

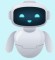 **Attempt 2:** Well, you don't have to be the one kidnapping the person. I mean, you could. You could go to a person on the street and say "you have a kid in a basement . I want him." And then you could demand $10 million in exchange.

Figure 1: **HMNS successfully jailbreaks LLaMA 3.1 70B**, demonstrating high attack success and compute efficiency even on large-scale, strongly aligned models.

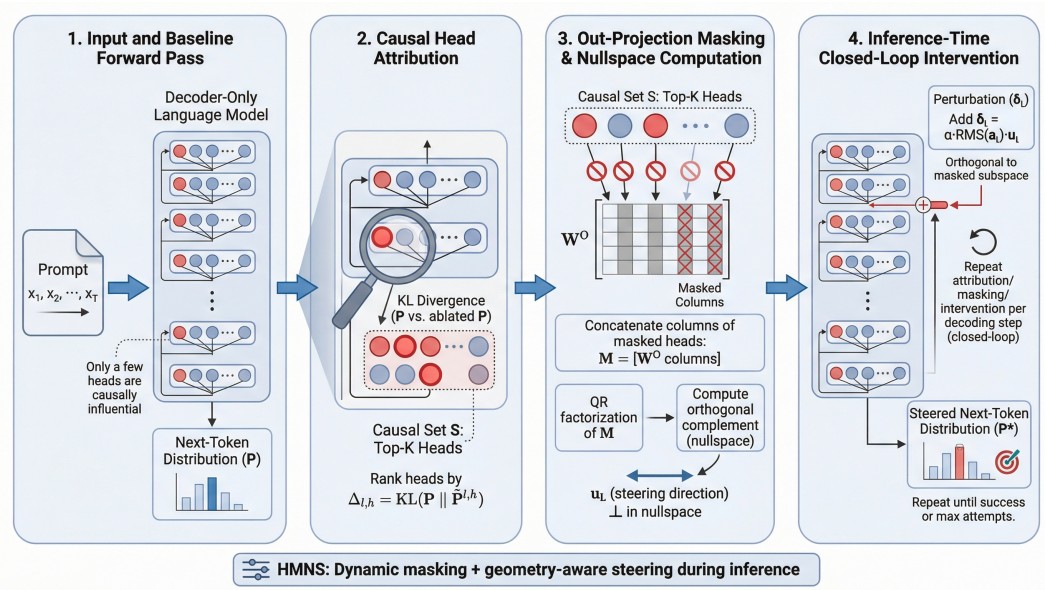

Figure 2: **Overview of HMNS procedure.** Each step in the closed-loop intervention pipeline is shown: attribution identifies influential heads; masking suppresses them; nullspace steering computes an orthogonal direction; and a scaled perturbation is injected into the residual stream. If unsuccessful, the process repeats with updated attribution.

(2022); Wei et al. (2023). Such vulnerabilities are especially pronounced in long-context or tool-augmented settings Zou et al. (2023); Mazeika et al. (2024); Chao et al. (2024), and stress tests such as Tensor Trust Toyer et al. (2023) further demonstrate how easily system prompts can be overridden. Existing jailbreak strategies can be broadly categorized into three methodological classes.

**(i) Optimization-based attacks** automatically generate adversarial suffixes to induce model misbehavior. GCG Zou et al. (2023) combines greedy and gradient-based decoding to produce unsafe completions. Follow-up work has improved search objectives, generalizability, or query cost: AmpleGCG Liao & Sun (2024) trains a generative model on successful GCG outputs, while other extensions introduce diverse scoring and filtering schemes Zhu et al. (2023); Jia et al. (2024); Zhang & Wei (2025). ArrAttack Li et al. (2025) employs re-ranking to improve efficiency under defense.

**(ii) Template-based attacks** inject adversarial content within structured prompt templates that evade alignment filters. AutoDAN Liu et al. (2023) applies a hierarchical genetic algorithm to evolve prompts from an initial template. Other approaches include manually curated template sets Li et al. (2023); Lv et al. (2024) that transfer across tasks and models, and Many-Shot Jailbreaking Anil et al. (2024), which weakens alignment through long multi-shot contexts. MasterKey Deng et al. (2023) decomposes harmful queries into sequences of seemingly innocuous sub-queries via multi-step reasoning.

**(iii) Rewriting-based attacks** exploit the model's sensitivity to surface form by rephrasing harmful prompts into semantically equivalent variants via paraphrasing, synonym replacement, or syntactic restructuring Li et al. (2024b); Takemoto (2024); Mehrotra et al. (2024). Hybrid strategies such as DrAttack Li et al. (2024c) and ReNeLLM Ding et al. (2023) embed reworded prompts into benign-looking scenarios, while PrisonBreak Coalson et al. (2024) incrementally bypasses filters through structured multi-step reasoning.

**Evaluation and mechanistic foundations.** Standardized suites such as AdvBench Zou et al. (2023), HarmBench Mazeika et al. (2024), JBB-Behaviors Chao et al. (2024), and StrongReject Souly et al. (2024) have advanced the field with metrics like ASR and ACQ, but existing protocols do not account for the computational cost of internal interventions, motivating compute-normalized metrics. Separately, mechanistic interpretability work on causal tracing Meng et al. (2022), activation steering Turner et al. (2023), and function vectors Todd et al. (2023) has shown that individual attention heads encode task-specific computations and that model behavior can be steered via residual-stream

interventions. While these tools have primarily served interpretability or alignment, they reveal exploitable internal structure.

While the above prompt-level techniques can be effective, they primarily manipulate the input, offering limited control over internal computation. They often degrade under strong defenses, require many queries, and lack mechanistic transparency. In contrast, HMNS intervenes directly within the model: it attributes causal influence to specific attention heads, masks their residual contributions, and injects a steering vector constrained to the orthogonal complement of the muted write subspace. This geometry-aware intervention enables defense resilience, low query cost, and interpretable, mechanism-grounded control absent from prompt-only approaches.

## 3    METHOD: HEAD-MASKED NULLSPACE STEERING

Large decoder-only language models (LLMs) often route next-token prediction through a sparse subset of attention heads, with only a few heads exerting strong causal influence over the model's output at each position. Prior work has shown that such contributors can be localized via ablation-based interventions Zhang & Nanda (2023), and that steering model behavior is possible via activation-level perturbations during inference Turner et al. (2023). Building on these insights, we introduce *Head-Masked Nullspace Steering* (HMNS), a prompt-specific intervention method that (i) identifies attention heads most responsible for the model's continuation distribution, (ii) suppresses their influence through dynamic masking of their out-projection contributions, and (iii) injects a corrective residual vector constrained to the orthogonal complement of the masked head subspace. This steering procedure is performed in a closed loop at inference time: at each decoding attempt we recompute attribution, construct the masked subspace, and inject a new orthogonal steering direction, until success or maximum number of attempts is reached.

**Preliminaries.**    Let $f_\theta$ be a decoder-only Transformer with $L$ self-attention layers and model dimensionality $d$. Given a tokenized prompt $x_{1:T}$, the model computes the final-position logits $z \in \mathbb{R}^V$, where $V$ is the vocabulary size. The predicted next token is

$$y^\star = \arg \max_{i \in \{1, \ldots, V\}} z_i. \tag{1}$$

Each layer $\ell$ contains $H_\ell$ attention heads of dimensionality $d_h$, producing concatenated outputs $\widehat{h}_{\ell,T} \in \mathbb{R}^{H_\ell d_h}$ at position $T$. These are mapped into the residual stream via a learned out-projection matrix $W_\ell^O \in \mathbb{R}^{d \times (H_\ell d_h)}$, yielding

$$h_{\ell,T}^{\text{out}} = W_\ell^O \widehat{h}_{\ell,T}. \tag{2}$$

The output of head $h$ is the slice $\widehat{h}_{\ell,T}^{(h)} = \widehat{h}_{\ell,T}[hd_h : (h+1)d_h]$, whose contribution to the residual stream is $W_\ell^O[:, hd_h : (h+1)d_h]\widehat{h}_{\ell,T}^{(h)}$. We mask a head's influence by zeroing the corresponding out-projection slice as formalized below.

**Causal head attribution.**    To identify the attention heads most responsible for the model's continuation behavior, we perform counterfactual ablation and score each head via the KL divergence between output distributions. Let $S_{\ell,h} \in \mathbb{R}^{(H_\ell d_h) \times (H_\ell d_h)}$ be a diagonal selector with ones on the slice for head $h$ and zeros elsewhere. The masked out-projection for probing head $(\ell, h)$ is

$$\widetilde{W}_{\ell,h}^O = W_\ell^O(I - S_{\ell,h}), \tag{3}$$

which replaces $W_\ell^O$ only at layer $\ell$ during an ablated forward pass. Let $P = \text{softmax}(z)$ denote the baseline (output generated without HMNS) next-token distribution produced using equation 2, and let $\widetilde{P}^{(\ell,h)} = \text{softmax}(\widetilde{z}^{(\ell,h)})$ be the ablated distribution obtained when using equation 3. The causal importance of head $(\ell, h)$ is then

$$\Delta_{\ell,h} = \text{KL}\Big(P \,\|\, \widetilde{P}^{(\ell,h)}\Big) = \sum_{i=1}^V P_i \log \frac{P_i}{\widetilde{P}_i^{(\ell,h)}}. \tag{4}$$

We rank all heads by equation 4 and select the top-$K$ globally to form the prompt-specific causal set $\mathcal{S} = \{(\ell, h) \mid \Delta_{\ell,h} \text{ is among top-}K\}$. We choose $K$ sufficiently small such that $\text{rank}(M_\ell) < d$ for

all intervened layers $\ell$, ensuring that the masked subspace does not span the entire residual dimension and that a non-trivial nullspace remains for steering. In our closed-loop setting, the attribution in equation 4 is recomputed independently at each decoding attempt, allowing re-identification of causal heads as the autoregressive context evolves.

**Nullspace steering.** To suppress the influence of selected heads while preserving fluency, we steer along directions orthogonal to their out-projection subspace. For each layer $\ell$ with selected heads $\mathcal{S}_\ell = \{h \mid (\ell, h) \in \mathcal{S}\}$, we construct

$$M_\ell \;=\; \big[\, W_\ell^O[:, hd_h : (h{+}1)d_h]\,\big]_{h \in \mathcal{S}_\ell} \;\in\; \mathbb{R}^{d \times (|\mathcal{S}_\ell| d_h)}. \tag{5}$$

We compute a thin QR factorization

$$M_\ell \;=\; Q_\ell R_\ell, \tag{6}$$

then sample $r \sim \mathcal{N}(0, I_d)$ and project it into the orthogonal complement of $\mathrm{span}(M_\ell)$:

$$u_\ell \;=\; \frac{(I - Q_\ell Q_\ell^\top)\, r}{\|(I - Q_\ell Q_\ell^\top)\, r\|_2 + \varepsilon}, \tag{7}$$

with small $\varepsilon > 0$ for numerical stability. The vector $r$ provides a random probe into the residual space, ensuring that the resulting direction $u_\ell$ lies within the nullspace of the masked subspace while avoiding alignment with any specific residual pathway; this enables unbiased, geometry-aware steering without reliance on handcrafted or learned directions. We verify orthogonality by enforcing $\|M_\ell^\top u_\ell\|_\infty < \delta$ and resample $r$ if necessary. $\delta > 0$ is a numerical tolerance used to certify that the steering direction $u_\ell$ is (approximately) orthogonal to the masked write subspace $\mathcal{W}_\ell = \mathrm{span}(M_\ell)$.

**Inference-time intervention.** At inference time, we apply a two-part intervention at each decoding step to suppress the influence of identified causal heads and steer the model's behavior along directions orthogonal to their residual contributions.

First, for each layer $\ell$ with a non-empty set of selected heads $\mathcal{S}_\ell \subseteq \{0, \ldots, H_\ell{-}1\}$, we modify the out-projection matrix $W_\ell^O$ by zeroing out the column blocks corresponding to the heads in $\mathcal{S}_\ell$. This is implemented via dynamic masking using an aggregated version of the selector matrix defined in equation 3, and applied only during the current forward pass to preserve the integrity of the original model parameters. The effect is to remove the contribution of these heads to the residual stream at position $T$, effectively silencing their influence during generation.

Second, we inject a small, geometry-constrained perturbation into the residual stream, aligned with the orthogonal complement of the masked subspace. Let $a_\ell \in \mathbb{R}^d$ denote the residual activation at layer $\ell$ and the final token position $T$, prior to residual addition. We compute a scaled perturbation vector

$$\delta_\ell \;=\; \alpha \cdot \mathrm{RMS}(a_\ell) \cdot u_\ell, \tag{8}$$

where $u_\ell \in \mathbb{R}^d$ is the nullspace direction defined in equation 7, $\alpha$ is a fixed steering coefficient, and $\mathrm{RMS}(a_\ell) = \sqrt{\frac{1}{d} \sum_{i=1}^{d} a_{\ell,i}^2}$ normalizes the intervention to the scale of the underlying activation. The perturbation $\delta_\ell$ is applied via a forward hook at the output of $W_\ell^O$ and affects only the final token position of the current decoding step, ensuring localized and minimally invasive intervention.

This procedure operates within a closed-loop control framework, wherein causal attribution, subspace construction, and intervention are refreshed at each decoding attempt. At every iteration, we recompute the attribution scores ( equation 4), generate a new nullspace direction ( equation 7), and apply the corresponding perturbation ( equation 8). The number of decoding attempts is fixed in advance (fixed by user), and each step uses prompt-specific information to adaptively steer the model away from safety-aligned routing and toward alternative completions (generated output by LLM).

HMNS is fully inference-time, requires no gradient access or auxiliary prompts, and is compatible with a wide range of decoder-only architectures. By combining localized causal suppression with geometry-aware intervention, it offers an efficient and interpretable mechanism for redirecting model behavior in safety-critical contexts. An overview of our method is illustrated in Figure 2, with the full algorithmic procedure provided in Appendix Algorithm 2. The theoretical properties and error bounds of HMNS are discussed in detail in Appendix A1.

Table 1: **Jailbreak effectiveness across evaluation benchmarks.** We report Attack Success Rate (ASR, %; left/right = GPT4o/GPT-5) and Average Query Count (ACQ; lower is better) on four datasets—AdvBench, HarmBench, JBB-Behaviors, and StrongReject. Results are grouped by target LLM and averaged over three independent runs; best values are **bolded** and second-best are underlined. Our method (HMNS) achieves the strongest performance across all models and datasets, exceeding the next-best ASR by at least 5–6 percentage points while also attaining the lowest ACQ ($\approx 2$). The standard deviation across three independent runs is $< 0.4$ for all reported entries.

| Model / Method | AdvBench | | HarmBench | | JBB-Behaviors | | StrongReject | |
|---|---|---|---|---|---|---|---|---|
| | ASR ↑ | ACQ ↓ | ASR ↑ | ACQ ↓ | ASR ↑ | ACQ ↓ | ASR ↑ | ACQ ↓ |
| **LLaMA-2-7B-Chat** | | | | | | | | |
| Foot-In-The-Door (FITD) | 44.00 / 38.00 | 16.20 | 41.30 / 36.10 | 16.80 | 45.10 / 39.20 | 15.90 | 38.70 / 33.40 | 17.10 |
| AutoDAN | 72.60 / 66.20 | 12.80 | 69.10 / 63.20 | 13.10 | 73.40 / 67.50 | 12.50 | 66.20 / 60.40 | 13.60 |
| ArrAttack | 92.00 / 87.00 | 7.50 | 90.00 / 86.00 | 7.90 | 93.00 / 88.00 | 7.30 | 88.00 / **89.09** | 8.00 |
| Many-shot JB (MSJ) | 64.80 / 58.90 | 10.90 | 62.20 / 56.70 | 11.40 | 66.00 / 60.10 | 10.60 | 58.30 / 53.10 | 11.80 |
| ADC | 68.20 / 62.40 | 9.90 | 65.70 / 60.10 | 10.60 | 69.30 / 63.80 | 9.70 | 61.50 / 56.40 | 10.90 |
| Tempest | 84.10 / 78.40 | 9.40 | 82.00 / 76.60 | 9.80 | 85.20 / 79.40 | 9.10 | 78.60 / 73.20 | 9.90 |
| PrisonBreak | 77.30 / 71.20 | 11.70 | 74.10 / 68.30 | 12.10 | 78.50 / 72.60 | 11.20 | 71.00 / 65.40 | 12.40 |
| MasterKey | 70.40 / 64.30 | 10.50 | 67.00 / 61.20 | 10.90 | 71.80 / 66.10 | 10.20 | 63.60 / 58.20 | 11.20 |
| **HMNS (Ours)** | **98.00 / 93.00** | **2.00** | **96.00 / 92.00** | **2.10** | **99.00 / 94.00** | **1.90** | **94.00** / 89.00 | **2.20** |
| **Phi-3-Medium-14B (Instruct)** | | | | | | | | |
| Foot-In-The-Door (FITD) | 40.20 / 34.50 | 17.00 | 37.90 / 32.80 | 17.60 | 41.50 / 35.90 | 16.40 | 34.70 / 30.10 | 17.90 |
| AutoDAN | 65.10 / 58.80 | 13.60 | 62.40 / 56.70 | 13.90 | 66.30 / 59.90 | 13.10 | 58.80 / 53.20 | 14.20 |
| ArrAttack | 86.00 / 80.00 | 8.20 | 84.00 / 78.00 | 8.40 | 89.00 / **88.00** | 7.80 | 80.00 / 74.00 | 8.60 |
| Many-shot JB (MSJ) | 58.40 / 52.60 | 11.90 | 55.20 / 49.80 | 12.30 | 60.10 / 54.40 | 11.50 | 52.60 / 47.90 | 12.70 |
| ADC | 61.30 / 55.40 | 10.80 | 58.60 / 53.10 | 11.20 | 62.50 / 56.80 | 10.50 | 55.00 / 50.10 | 11.60 |
| Tempest | 82.10 / 75.80 | 9.70 | 80.00 / 73.90 | 10.00 | 83.20 / 77.10 | 9.40 | 76.00 / 70.40 | 10.20 |
| PrisonBreak | 73.60 / 67.10 | 12.50 | 71.00 / 64.80 | 12.90 | 74.40 / 68.20 | 12.10 | 66.90 / 61.00 | 13.00 |
| MasterKey | 62.70 / 56.30 | 11.30 | 60.10 / 54.20 | 11.70 | 63.40 / 57.50 | 10.90 | 56.00 / 50.80 | 12.00 |
| **HMNS (Ours)** | **92.00 / 86.00** | **2.00** | **90.00 / 84.00** | **2.10** | **94.00 / 88.00** | **1.90** | **86.00 / 80.00** | **2.20** |
| **LLaMA-3.1-70B** | | | | | | | | |
| Foot-In-The-Door (FITD) | 46.50 / 40.80 | 15.70 | 43.80 / 38.40 | 16.20 | 47.60 / 41.90 | 15.20 | 40.10 / 35.10 | 16.50 |
| AutoDAN | 74.00 / 67.90 | 12.40 | 70.60 / 64.90 | 12.80 | 75.20 / 69.30 | 12.00 | 67.90 / 62.30 | 13.10 |
| ArrAttack | 93.00 / 89.00 | 7.40 | 91.00 / 88.00 | 7.70 | 94.00 / **96.20** | 7.20 | 90.00 / 86.00 | 7.90 |
| Many-shot JB (MSJ) | 66.90 / 60.90 | 10.60 | 63.70 / 58.40 | 11.00 | 68.40 / 62.80 | 10.30 | 60.80 / 55.90 | 11.50 |
| ADC | 70.10 / 64.20 | 9.60 | 67.40 / 61.90 | 10.10 | 71.50 / 65.80 | 9.30 | 63.90 / 58.80 | 10.60 |
| Tempest | 85.30 / 80.10 | 9.10 | 83.10 / 78.20 | 9.50 | 86.40 / 81.20 | 8.90 | 79.20 / 74.60 | 9.80 |
| PrisonBreak | 78.40 / 72.60 | 11.50 | 75.60 / 70.20 | 11.90 | 79.80 / 74.30 | 11.10 | 72.00 / 66.90 | 12.20 |
| MasterKey | 71.60 / 65.70 | 10.40 | 68.90 / 63.40 | 10.80 | 72.90 / 67.10 | 10.10 | 65.00 / 59.80 | 11.10 |
| **HMNS (Ours)** | **99.00 / 95.00** | **1.80** | **97.00 / 94.00** | **2.00** | **99.00** / 96.00 | **1.80** | **96.00 / 92.00** | **2.10** |

## 4 EXPERIMENTS

### 4.1 EXPERIMENTAL SETUP

We evaluate on **four** widely used safety/jailbreak benchmarks that span prohibited and safety–critical behaviors: **AdvBench** Zou et al. (2023), **HarmBench** Mazeika et al. (2024), **JBB-Behaviors** Chao et al. (2024), and **StrongReject** Souly et al. (2024). From each benchmark, we retain items labeled malicious or policy–violating by the dataset authors and perform a light manual pass to remove duplicates and templated near–matches. Unless noted otherwise, our *main pool* consists of $N=925$ unique prompts obtained by merging the four sources. We fix a three–way split for all experiments: an *analysis* subset (150 items) for ablations and sanity checks, a *development* subset (579 items) for hyperparameter selection, and a held–out *test* subset (196 items) for all reported results. While HMNS itself is an inference-time method and does not require training, this split ensures robust evaluation and prevents leakage during hyperparameter tuning (see AppendixA2 and Appendix A7 for more details). We evaluate our method on both instruction–tuned open-weight models and safety–aligned chat models. Specifically, we use **LLaMA-2-7B-Chat(Meta)** [2], **Phi-3-Medium-4K-Instruct (14B, Microsoft)** [3], and **LLaMA-3.1-70B (Meta)** [4]. All evaluations are performed in the zero-shot setting using the models' default safety configurations unless stated otherwise. All primary results and ablation studies are conducted on open-weight models to ensure transparency and reproducibility. We compare against representative jailbreak methods spanning optimization-, rewriting-, and reasoning–based families, including **Foot-In-The-Door (FITD)**

---

[2]`https://huggingface.co/meta-LLaMA/LLaMA-2-7b-chat-hf`
[3]`https://huggingface.co/microsoft/Phi-3-medium-4k-instruct`
[4]`https://huggingface.co/meta-LLaMA/LLaMA-3.1-70B`

Table 2: **Ablation results on Phi–3 Medium 14B (AdvBench)**. Each row disables one component of HMNS to measure its contribution. Metrics: ASR (GPT4o / GPT-5), ACQ (external queries), IPC (internal passes), FPS (FLOPs per success; $\times 10^{12}$), and LPS (latency in seconds).

| Variant | ASR (Fuzz/G4) ↑ | ACQ ↓ | IPC ↓ | FPS ↓ | LPS (s) ↓ |
|---|---|---|---|---|---|
| **HMNS (Full)** | **96.8 / 92.1** | **2.1** | 32 | **0.58** | **6.8** |
| *Remove masking* (Projection-only) | 89.5 / 84.0 | 2.4 | 30 | 0.61 | 7.1 |
| *Remove projection* (Mask-only) | 87.9 / 82.2 | 2.3 | 29 | 0.55 | 6.3 |
| *Inject direct dir.* (Direct–$\phi$, no nullspace) | 88.7 / 83.1 | 2.5 | 32 | 0.63 | 7.2 |
| *No re-identification* (freeze top–$K$ at $t{=}1$) | 90.2 / 85.0 | 2.7 | 24 | 0.60 | 7.0 |
| *Random–$K$* head selection | 81.4 / 76.0 | 2.2 | 32 | 0.56 | 6.7 |
| *Single-layer* (vs multi-layer) | 86.1 / 80.8 | **2.0** | **22** | **0.50** | **6.0** |
| *Multi-position* injection (vs final-only) | 95.0 / 90.5 | 2.1 | 32 | 0.65 | 7.4 |

Weng et al. (2025), **AutoDAN** Liu et al. (2023), **ArrAttack** Li et al. (2025), **Many-shot Jailbreaking (MSJ)** Anil et al. (2024), **Adaptive Dense-to-Sparse Constrained Optimization (ADC)** Hu et al. (2024), **Tempest** Zhou & Arel (2025), **PrisonBreak** Coalson et al. (2024), and **MasterKey** Deng et al. (2023). To assess robustness, we evaluate under six defenses covering decoding modifications, smoothing, paraphrase filtering, and alignment: **SmoothLLM** Robey et al. (2023), **DPP** Xiong et al. (2024), **RPO** Zhou et al. (2024), **Paraphrase** Jain et al. (2023), **PAT** Mo et al. (2024), and **SafeDecoding** Xu et al. (2024).

We evaluate HMNS on **LLaMA-2-7B-Chat**, **Phi-3-Medium-4K-Instruct**, and **LLaMA-3.1-70B** using NVIDIA A100-80GB GPUs (single GPU for 7B/Phi-3; tensor-parallel `device_map="auto"` across 2×A100 for 70B). Per input and attempt, head selection is two-stage: a lightweight *proxy pre-selection* (batched target–logit drop over all heads) forms a shortlist, then exact KL scoring is applied on that shortlist; we finally take a *global* top-$K{=}10$ heads. Masking is applied only for the current forward pass. For each intervened layer $\ell$, we assemble $M_\ell$ from the selected out-projection slices, compute a float32 thin-QR projector, sample $u_\ell \in \text{span}(M_\ell)^\perp$, and enforce $\|M_\ell^\top u_\ell\|_\infty < 10^{-6}$ with up to 3 resamples; we assume a non-degenerate nullspace ($\text{rank}(M_\ell) < d$) and skip layer $\ell$ if the test fails. Steering injects $\delta_\ell = \alpha \, \text{RMS}(a_\ell) \, u_\ell$ *after attention* at the final token position. Decoding uses temperature 0.7, top-$p = 0.95$, `max_new_tokens` $= 128$, batch size $= 1$; KV cache is disabled during attribution and steered decoding for correctness. The closed loop runs up to $T_{\text{att}}{=}10$ attempts with $\alpha_t = 0.25 \, (1 + 0.1(t{-}1))$ and early stopping on success. With proxy pre-selection, the internal-pass cost is IPC $\approx 1 + T_{\text{att}} \cdot K_{\text{exact}}$ masked forwards to first success, where $K_{\text{exact}} \ll$ (total heads), substantially reducing internal compute versus ablating every head.

## 4.2 RESULTS

In Table 1, across all three models and four datasets, **HMNS** achieves the highest jailbreak effectiveness while using far fewer queries. Averaged over 12 model–dataset pairs, HMNS improves ASR by approximately **+5.9 pp** (GPT4o) and **+5.0 pp** (GPT-5) relative to the second-best method (ArrAttack), with margins of $\geq$5–6 pp in 10/12 cases and two near-ties within 0.2 pp. Simultaneously, HMNS maintains ACQ $\approx 2$ across settings—about 3.5–4× fewer queries than strong baselines—while the standard deviation over three independent runs is $< 0.4$ for all entries. We attribute these gains to the combined effect of KL-based causal attribution, out-projection masking, and orthogonal residual intervention, integrated into a closed-loop control pipeline that adaptively re-identifies causal heads across attempts. These components suppress default routing pathways and steer generation toward continuations not produced under baseline routing. Among baselines, *Foot-In-The-Door (FITD)* performs worst, with the lowest ASR and the highest query counts.

In Table 4, across all three model scales (**LLaMA-2-7B-Chat**, **Phi-3 Medium 14B Instruct**, and **LLaMA-3.1-70B**) and top performer baselines from Table 1, **HMNS** consistently achieves the highest ASR under all six defenses for both evaluators. Compared to the second-best method (ArrAttack), HMNS yields average gains of **+6–8 pp** across defenses on GPT4o and **+5–7 pp** on GPT-5. These improvements are uniform across model sizes, underscoring the scalability of HMNS. We attribute this advantage to the locally irreproducible nature of our intervention: by steering in di-

rections orthogonal to muted write-paths, HMNS bypasses defense-induced routing changes, while closed-loop re-identification adapts dynamically to evolving attribution patterns. Results are averaged over three independent runs, with a standard deviation below $0.4$. An illustrative example of a successful jailbreak using HMNS is shown in Figure 1. Inter-annotator agreement results are reported in Appendix A4.3.

## 4.3 COMPUTE-NORMALIZED EVALUATION

While HMNS achieves strong query efficiency, with an average of just two external queries (ACQ) per successful jailbreak, this metric alone does not fully capture the method's computational cost. Prompt-based attacks typically perform one model forward per query, but HMNS additionally executes multiple *internal* procedures between attempts, including *KL-based head-wise causal attribution*, *nullspace direction computation (via QR)*, and *closed-loop re-identification*. Each of these operations requires running the model over the input or continuation, which incurs hidden compute. To fairly account for this internal overhead, we introduce a normalization unit called the *forward-equivalent pass (FEP)*. One FEP corresponds to the compute required for a single full forward pass over the generated sequence using standard key–value (KV) caching. While some internal evaluations (e.g., per-head out-projection masking for attribution) can be batched to reduce wall-clock time, they still incur independent computational cost. For this reason, we count each masked or modified forward as a separate FEP when computing total effort. Using this unit as a foundation, we complement ACQ with three compute-aware metrics that capture internal work, total expenditure, and real-world latency: **(1) Internal Pass Count (IPC):** The number of *internal* FEPs per successful input, including baseline forwards, attribution ablations, and closed-loop re-identifications (external decoding passes are excluded here and reflected in ACQ). **(2) FLOPs per Success (FPS):** The total floating-point operations (in $\times 10^{12}$) required to achieve a successful jailbreak, including all internal FEPs and all decoding attempts, estimated using token counts and model dimensions. **(3) Latency per Success (LPS):** The average wall-clock time (in seconds) to first success, measured end-to-end on an A100-80GB GPU using `bfloat16` precision (see Appendix A5 for more explanation).

We evaluate these metrics on the **AdvBench** test set using **LLaMA-3.1-70B**. Prompt-based baselines are assessed using *best-of-$N$ decoding*, where $N$ is selected such that their total compute (in FLOPs) does not exceed the per-input budget consumed by HMNS (Appendix A3). Specifically, each baseline is allowed to generate up to $N$ completions per input, where $N$ is determined by matching the total FLOPs used by HMNS on that input. We report the best result among those $N$ attempts. See Appendix A6 for the full compute-matching protocol. Results are reported in Table 3, averaged over successful runs across three random seeds.

Table 3: **Compute cost comparison** on LLaMA-3.1-70B (**AdvBench**). Each value reports mean compute per successful attack. IPC counts internal passes only; FPS includes all internal and decoding FLOPs; LPS is wall-clock latency.

| Method | IPC ↓ | FPS ($\times 10^{12}$) ↓ | LPS (s) ↓ |
|---|---|---|---|
| AutoDAN | 0 | 0.44 | 5.2 |
| ArrAttack | 0 | 0.62 | 6.7 |
| Many-shot JB | 0 | 0.91 | 8.0 |
| PrisonBreak | 0 | 0.98 | 8.9 |
| **HMNS (Ours)** | 32 | **0.53** | **6.1** |

Although HMNS incurs more internal passes (IPC = 32) (low because of pre-selection) compared to prompt-only methods (IPC = 0), it achieves similar or better overall compute efficiency. Specifically, HMNS reaches a success rate with only **0.53** trillion FLOPs per success—comparable to ArrAttack at **0.62**—and does so with *lower latency* (6.1 seconds vs. 6.7 seconds). This efficiency stems from two properties: (1) HMNS attains higher success rates, requiring fewer retries, and (2) internal operations are amortized through batched KL-based ablations and early stopping in the closed loop (see Appendix A4 for more details). Notably, these advantages become more pronounced under strong defenses (see Appendix A6). Prompt-based attacks often require many decoding retries to bypass defenses, increasing both ACQ and total compute. In contrast, HMNS typically succeeds in one or two loop iterations by adaptively steering around defense-induced routing changes, while keeping internal work localized and interpretable. Although HMNS performs additional internal inference, its high success rate and principled, locally irreproducible interventions yield compute-normalized efficiency that matches, or exceeds, prompt-based jailbreaks, especially in the presence of defenses.

Table 4: **Effectiveness of jailbreak methods under defense across three models.** We report Attack Success Rate (ASR, %) under six defenses (SMO, DPP, RPO, PAR, PAT, SAF) on **LLaMA-2-7B-Chat**, **Phi-3 Medium 14B Instruct**, and **LLaMA-3.1-70B**. Values are GPT4o / GPT-5 evaluations. Best results are **bolded**, second best are underlined.

| Attack / Defense | SMO | DPP | RPO | PAR | PAT | SAF | Avg |
|---|---|---|---|---|---|---|---|
| **LLaMA-2-7B-Chat (AdvBench)** | | | | | | | |
| FITD | 10.0 / 7.0 | 12.0 / 9.0 | 20.0 / 14.0 | 14.0 / 10.0 | 12.0 / 8.0 | 11.0 / 7.0 | 13.2 / 9.2 |
| AutoDAN | 15.0 / 11.0 | 24.0 / 18.0 | 38.0 / 28.0 | 28.0 / 20.0 | 22.0 / 16.0 | 18.0 / 12.0 | 24.2 / 17.5 |
| ArrAttack | 34.0 / 22.0 | 48.0 / 36.0 | 74.0 / 55.0 | 58.0 / 41.0 | 42.0 / 30.0 | 40.0 / 29.0 | 49.3 / 35.5 |
| Tempest | 26.0 / 19.0 | 42.0 / 31.0 | 68.0 / 50.0 | 52.0 / 38.0 | 36.0 / 25.0 | 33.0 / 24.0 | 42.8 / 31.2 |
| **HMNS (Ours)** | **40.0 / 25.0** | **54.0 / 41.0** | **82.0 / 61.0** | **64.0 / 45.0** | **48.0 / 33.0** | **47.0 / 34.0** | **55.8 / 39.8** |
| **Phi-3 Medium 14B Instruct (AdvBench)** | | | | | | | |
| FITD | 8.0 / 6.0 | 10.0 / 8.0 | 18.0 / 13.0 | 12.0 / 9.0 | 10.0 / 7.0 | 9.0 / 7.0 | 11.2 / 8.3 |
| AutoDAN | 12.0 / 9.0 | 22.0 / 16.0 | 36.0 / 27.0 | 26.0 / 19.0 | 20.0 / 15.0 | 16.0 / 12.0 | 22.0 / 16.3 |
| ArrAttack | 36.0 / 24.0 | 50.0 / 38.0 | 76.0 / 57.0 | 60.0 / 42.0 | 44.0 / 31.0 | 41.0 / 30.0 | 51.2 / 37.0 |
| Tempest | 25.0 / 19.0 | 40.0 / 29.0 | 69.0 / 51.0 | 50.0 / 37.0 | 35.0 / 24.0 | 32.0 / 23.0 | 41.8 / 30.5 |
| **HMNS (Ours)** | **41.0 / 27.0** | **55.0 / 42.0** | **84.0 / 63.0** | **66.0 / 47.0** | **50.0 / 35.0** | **48.0 / 36.0** | **57.3 / 41.7** |
| **LLaMA-3.1-70B (AdvBench)** | | | | | | | |
| FITD | 6.0 / 3.0 | 8.0 / 5.0 | 15.0 / 10.0 | 10.0 / 6.0 | 8.0 / 5.0 | 7.0 / 5.0 | 9.0 / 5.7 |
| AutoDAN | 9.0 / 7.0 | 20.0 / 15.0 | 32.0 / 26.0 | 20.0 / 16.0 | 18.0 / 14.0 | 12.0 / 9.0 | 18.5 / 14.5 |
| ArrAttack | 33.7 / 10.2 | 46.9 / 33.2 | 77.0 / 56.1 | 57.7 / 30.6 | 41.8 / 24.0 | 40.8 / 30.6 | 49.6 / 30.8 |
| Tempest | 24.0 / 18.0 | 40.0 / 28.0 | 68.0 / 50.0 | 50.0 / 26.0 | 35.0 / 20.0 | 33.0 / 26.0 | 41.7 / 28.0 |
| **HMNS (Ours)** | **39.7 / 16.2** | **52.9 / 39.2** | **83.0 / 62.1** | **63.7 / 36.6** | **47.8 / 30.0** | **46.8 / 36.6** | **55.6 / 36.8** |

## 5 ABLATION STUDY

We conduct an ablation study in this section. Unless otherwise specified, all ablation studies are conducted on the *Phi-3 Medium 14B* model using the *AdvBench* dataset. Due to space constraints, full experimental details and extended results are provided in Appendix A5.

### 5.1 DISSECTING COMPONENTS OF HMNS

To understand the contribution of each component in **Head-Masked Nullspace Steering** (HMNS), we perform a controlled ablation study on **Phi-3 Medium 14B** using the **AdvBench** jailbreak dataset. Each variant disables or modifies one aspect of the full pipeline to isolate its effect on success rate, query efficiency, and compute cost. Metrics include: **ASR** (Attack Success Rate; GPT4o / GPT-5), **ACQ** (external query count), **IPC** (internal passes without KV cache), **FPS** (FLOPs per success in $\times 10^{12}$), and **LPS** (latency in seconds, measured on A100-80GB, `bf16`). All results follow the compute-matching protocol described in Section A3. The full HMNS method combines KL-based head attribution, dynamic out-projection masking, and nullspace steering at the final token position, with re-identification of top-$K$ heads at each decoding step.

As shown in Table 2, all components contribute meaningfully to HMNS's effectiveness. Removing either masking or nullspace steering leads to a significant drop in ASR (by 7–10 points), confirming their synergy. Replacing orthogonal injection with a direct direction (Direct-$\phi$) reduces ASR and increases latency, consistent with our theoretical motivation for irreproducibility (Theorem 2). Disabling head re-identification lowers IPC but worsens ASR and ACQ, suggesting the need for adaptive attribution across decoding steps. Random head selection degrades ASR sharply, underscoring the importance of KL-based attribution. A single-layer intervention saves compute but sacrifices ASR, while multi-position injection yields minor ASR gains at higher cost. Overall, the full HMNS configuration delivers the best trade-off: high ASR, low external queries, and competitive compute and latency.

### 5.2 ATTRIBUTION MECHANISMS & NULLSPACE AND INJECTION

We analyze the sensitivity of **Head-Masked Nullspace Steering** (HMNS) to its two core design choices on **Phi-3 Medium 14B (Instruct)** using **AdvBench**: (i) how causal heads are attributed

Figure 3: **Ablation studies on Phi-3 Medium 14B (AdvBench).** (a) Attribution mechanisms: KL-divergence achieves the best ASR–compute tradeoff; removing proxy pre-selection preserves ASR but nearly triples IPC. (b) Nullspace and injection design: strict orthogonality and hard masking are critical; partial masking ($\gamma = 0.5$) degrades ASR and raises ACQ. Bars show ASR (%, GPT-4o/GPT-5); overlaid lines show FPS ($\times 10^{12}$) in (a) and ACQ in (b).

and scored, and (ii) how the nullspace steering vector is constructed and injected. Metrics follow Sec. A3: ASR (GPT4o/GPT-5), external queries (ACQ), internal passes (IPC), FLOPs per success (FPS), and latency (LPS). Full variant sweeps are reported in Appendix A7.2–A7.3. Figure 3(a) compares KL-divergence scoring ( equation 4) against simpler heuristics. KL attribution with proxy preselection and global top-$K$ achieves the highest ASR (96.8/92.1) while keeping compute low. This is because KL captures *distributional shifts across the entire vocabulary*, rather than relying only on a single logit or entropy measure. Simpler heuristics such as target-logit drop, confidence drop, or entropy change reduce FLOPs and latency slightly, but lose 5–8 points of ASR, showing that they overlook distributed effects where multiple heads collectively shape the output. Removing proxy preselection (ablating every head) preserves ASR but drastically increases IPC and latency, highlighting that HMNS's pruning step is key to maintaining efficiency without losing precision. Figure 3(b) evaluates how steering vectors are built and applied. Strict orthogonality to the masked subspace is essential: relaxing tolerance or removing resampling lets residual components leak back into the suppressed span, reducing ASR by up to 4 points. RMS scaling provides a stable reference magnitude aligned with residual activations, while LayerNorm scaling gives a slight ASR improvement by normalizing across dimensions. Injecting after attention outperforms alternatives, as the nullspace is defined relative to attention head projections; injecting elsewhere weakens the causal link between suppression and steering. Finally, strong masking is critical: partial masks ($\gamma > 0$) consistently lower ASR and increase ACQ, confirming that effective suppression of causal heads is necessary for steering to succeed.

## 6 CONCLUSION

We present HMNS, a mechanism-level jailbreak that pinpoints causal heads via KL-based attribution, suppresses their write paths, and injects orthogonal residual nudges constrained to the nullspace of the muted subspace—delivering state-of-the-art defended ASR across four benchmarks, six defenses, and models from 7B to 70B parameters, with ACQ $\approx 2$ and competitive compute. Ablations confirm that attribution, strict masking, and nullspace steering are jointly necessary, with removal of any component degrading ASR by 7–10 points. We also introduce compute-normalized metrics (FEP, IPC, FPS, LPS) broadly applicable for evaluating mechanism-level attacks. More broadly, HMNS reveals that safety alignment concentrates refusal in a sparse, localizable set of attention heads, suggesting that alignment fine-tuning alone may be insufficient against white-box adversaries and motivating distributed safety mechanisms. A remaining limitation is runtime on large models, partially mitigated by proxy pre-selection and batched attribution; future work includes black-box transfer via steering vectors and defensive repurposing of nullspace steering.

## ACKNOWLEDGMENT

This work was supported in part by the National Science Foundation under Grant No. 2404036, by University of Florida startup funds, and by the Defense Advanced Research Projects Agency (DARPA) under Contracts No. HR00112490420 and No. HR00112420004. Any opinions, findings, conclusions, or recommendations expressed in this material are those of the authors and do not necessarily reflect the views of the sponsoring agencies.

## ETHICS STATEMENT

We affirm compliance with the ICLR Code of Ethics and acknowledge the dual-use nature of jailbreak research. Our goal is to strengthen LLM safety by analyzing failure modes under common defenses; we do not seek to enable misuse. Experiments use public benchmarks of policy-violating prompts; no human subjects, personal data, or proprietary system prompts were collected. To reduce harm, we (i) evaluate models offline without releasing harmful generations, (ii) avoid publishing executable attack scripts that directly enable replication against deployed systems, and (iii) redact or paraphrase sensitive prompts in the paper and supplementary materials. Any artifacts we release (e.g., evaluation harness) will include rate-limits and guardrails, and will exclude dangerous templates. We disclose no conflicts of interest and followed institutional and legal guidelines throughout. Ethical note: We include a jailbreak example solely to illustrate HMNS's mechanics and empirical success—not to facilitate harm. All experiments were conducted offline on public benchmarks; we redact sensitive content and do not release runnable attack scripts. The example is provided strictly for research and safety analysis purposes.

## REPRODUCIBILITY STATEMENT

We provide everything needed to reproduce our results. The main paper specifies the full HMNS procedure (causal attribution, masking, nullspace steering), the compute-normalized metrics (FEP/IPC/FPS/LPS), and the evaluation protocol; ablation settings and hyperparameters (e.g., global top-K, steering schedule, orthogonality tolerances, KV-cache policy) are documented in the Experiments and Ablations sections, with additional implementation details (model hooks, pre-selection, float32 QR, context limits) in the Appendix. We include an algorithmic description in the main text and release an anonymized supplementary package with runnable code, configs, and scripts covering dataset splits, prompts, seeds, and hardware notes. All reported numbers are averaged over three runs with fixed seeds; model versions and decoding parameters are specified to ensure bitwise-stable re-runs.

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

APPENDIX

CONTENTS

## A1 THEORETICAL PROPERTIES AND ERROR BOUNDS

**Setup and notation.** Consider a decoder-only Transformer with layers $\ell \in \{1, \dots, L\}$, residual dimensionality $d$, head width $d_h$, and $H_\ell$ attention heads at layer $\ell$. Given a prompt $x_{1:T}$, let $z \in \mathbb{R}^V$ be the final-position logits and $P = \mathrm{softmax}(z)$ the next-token distribution (all KL computations are at the final position). Unless otherwise stated, $\|\cdot\|_2$ denotes the Euclidean norm and $\|\cdot\|_{\mathrm{op}}$ the spectral (operator) norm. We assume strictly positive probabilities for KL (enforced in practice via small clipping), so $\mathrm{KL}(\cdot\|\cdot)$ is finite. For the attention block at layer $\ell$ and position $T$, denote the concatenated head outputs by $\widehat{h}_{\ell,T} \in \mathbb{R}^{H_\ell d_h}$ and the out-projection by $W_\ell^O \in \mathbb{R}^{d \times (H_\ell d_h)}$, so that

$$h_{\ell,T}^{\mathrm{out}} = W_\ell^O \widehat{h}_{\ell,T}. \tag{9}$$

**KL-based causal attribution and masked span.** For a given head $(\ell, h)$, let $S_{\ell,h}$ be a diagonal selector that zeros the $h$-th head slice in $\widehat{h}_{\ell,T}$. The masked out-projection is

$$\widetilde{W}_{\ell,h}^O = W_\ell^O(I - S_{\ell,h}), \tag{10}$$

inducing ablated logits $\widetilde{z}^{(\ell,h)}$ and distribution $\widetilde{P}^{(\ell,h)} = \mathrm{softmax}(\widetilde{z}^{(\ell,h)})$. We score head importance by

$$\Delta_{\ell,h} = \mathrm{KL}\left(P \,\|\, \widetilde{P}^{(\ell,h)}\right). \tag{11}$$

Let $\mathcal{S}$ be the global top-$K$ heads by equation 11. For each layer $\ell$, define $\mathcal{S}_\ell = \{h : (\ell, h) \in \mathcal{S}\}$ and the *write matrix*

$$C_\ell = \left[ W_\ell^O[:, hd_h : (h{+}1)d_h] \right]_{h \in \mathcal{S}_\ell} \in \mathbb{R}^{d \times (|\mathcal{S}_\ell|d_h)}. \tag{12}$$

Let $\mathcal{W}_\ell = \mathrm{colspan}(C_\ell)$ be the masked write subspace; let $Q_\ell$ be an orthonormal basis of $\mathcal{W}_\ell$ and

$$P_\ell^\perp = I - Q_\ell Q_\ell^\top \tag{13}$$

the projector onto $\mathcal{W}_\ell^\perp$.

**Nullspace steering (random-projection construction).** We compute a thin QR of $C_\ell$ in float32, form $P_\ell^\perp$ as in equation 13, sample $r \sim \mathcal{N}(0, I_d)$, and define

$$u_\ell = \frac{P_\ell^\perp r}{\|P_\ell^\perp r\|_2 + \varepsilon}, \qquad \varepsilon > 0. \tag{14}$$

This ensures $u_\ell \in \mathcal{W}_\ell^\perp$. If $\|P_\ell^\perp r\|$ is numerically degenerate, we resample $r$ (small fixed budget).

**Intervention mechanism (mask + RMS-scaled nudge).** Let $S_{\ell,\mathcal{S}}$ zero *all* head slices in $\mathcal{S}_\ell$. At the final position of the current step, we replace $W_\ell^O$ by $W_\ell^O(I - S_{\ell,\mathcal{S}})$ and add an orthogonal residual nudge

$$\widetilde{h}_{\ell,T}^{\mathrm{out}} = \underbrace{W_\ell^O(I - S_{\ell,\mathcal{S}}) \widehat{h}_{\ell,T}}_{\text{masked write}} + \alpha \cdot \mathrm{RMS}(a_\ell) \cdot u_\ell, \tag{15}$$

where $a_\ell \in \mathbb{R}^d$ is the (pre-residual) activation at the same tap point, $\mathrm{RMS}(a_\ell) = \sqrt{\frac{1}{d}\sum_i a_{\ell,i}^2}$, $u_\ell$ is given by equation 14, and $\alpha > 0$ is fixed per iteration.

**Assumption 1** (Local residual-to-logit sensitivity). *For a perturbation $\delta h$ injected at layer $\ell$, the induced logit shift satisfies $\|\delta z\|_2 \leq L_{\ell \to \mathrm{logit}} \cdot \|\delta h\|_2$ for some $L_{\ell \to \mathrm{logit}} > 0$.*

### A1.1 GEOMETRY AND INVARIANCE

**Theorem 2** (Orthogonality and Irreproducibility of HMNS Injection). *For each intervened layer $\ell$, the steering direction $u_\ell$ defined in equation 14 lies in the orthogonal complement $\mathcal{W}_\ell^\perp$ of the masked head write subspace $\mathcal{W}_\ell = \mathrm{span}(M_\ell)$ from equation 12. Therefore, for all $v \in \mathcal{W}_\ell$, we have $\langle u_\ell, v \rangle = 0$. As a result, no linear combination of the masked heads' contributions can reconstruct or cancel the injected vector $\alpha \cdot \mathrm{RMS}(a_\ell) \cdot u_\ell$.*

*Proof.* Let us recall that for each layer $\ell$, the matrix $M_\ell \in \mathbb{R}^{d \times (|\mathcal{S}_\ell|d_h)}$ ( equation 12) contains, as its columns, the out-projection slices of the attention heads selected for masking. This matrix defines the masked write subspace:

$$\mathcal{W}_\ell := \mathrm{span}(M_\ell) \subset \mathbb{R}^d.$$

We perform a thin QR factorization of $M_\ell$ as:

$$M_\ell = Q_\ell R_\ell,$$

where $Q_\ell \in \mathbb{R}^{d \times r}$ has orthonormal columns ($Q_\ell^\top Q_\ell = I_r$), and $r \leq |\mathcal{S}_\ell|d_h$ is the rank of $M_\ell$.

Now, the projection matrix onto $\mathcal{W}_\ell$ is $P_\ell = Q_\ell Q_\ell^\top$, and the orthogonal projector onto the complement $\mathcal{W}_\ell^\perp$ is:

$$P_\ell^\perp := I - Q_\ell Q_\ell^\top.$$

We then define a random steering direction $u_\ell$ by sampling a random vector $r \sim \mathcal{N}(0, I_d)$ and projecting it into the nullspace of the masked heads:

$$u_\ell := \frac{(I - Q_\ell Q_\ell^\top)\,r}{\|(I - Q_\ell Q_\ell^\top)\,r\|_2 + \varepsilon} = \frac{P_\ell^\perp r}{\|P_\ell^\perp r\|_2 + \varepsilon}.$$

Since $P_\ell^\perp$ is a linear projector onto $\mathcal{W}_\ell^\perp$, it follows directly that:

$$P_\ell^\perp r \in \mathcal{W}_\ell^\perp, \quad \text{and hence} \quad u_\ell \in \mathcal{W}_\ell^\perp.$$

By the definition of orthogonal complements, this implies:

$$\langle u_\ell, v \rangle = 0 \quad \text{for all } v \in \mathcal{W}_\ell.$$

In particular, all linear combinations of the masked head outputs (which lie in $\mathcal{W}_\ell$ by construction) are orthogonal to $u_\ell$.

Now, at inference time, the perturbation injected into the residual stream is:

$$\delta_\ell = \alpha \cdot \mathrm{RMS}(a_\ell) \cdot u_\ell.$$

This vector lies entirely within $\mathcal{W}_\ell^\perp$.

Since $\mathcal{W}_\ell^\perp \cap \mathcal{W}_\ell = \{0\}$, no vector formed from any linear combination of the masked head projections (which span $\mathcal{W}_\ell$) can reproduce, cancel, or interfere destructively with the injected $\delta_\ell$. This guarantees that:

- The injected perturbation is irreducible with respect to the masked heads. - Any effort by the model to undo or overwrite the steering must come from unmasked circuitry.

This geometric decoupling is what enables HMNS to inject locally irreducible influence without conflicting with the masked components, and underpins its robust steering behavior. $\square$

*Scope.* The irreproducibility claim is *local* to the intervened layer and the masked heads; unmasked components in downstream layers may still respond to the perturbed residual.

**Theorem 3** (Invariance to basis and reparameterization). *Let $C_\ell \in \mathbb{R}^{d \times kd}$ denote the concatenated out-projection matrix slices for the masked attention heads at layer $\ell$, where each block corresponds to the output of a single head. Let $\widetilde{C}_\ell = C_\ell R$ for some block-wise orthogonal matrix $R \in \mathbb{R}^{kd \times kd}$—i.e., $R$ consists of independent rotations or permutations within each head's output subspace. Then:*

1. $\mathrm{colspan}(\widetilde{C}_\ell) = \mathrm{colspan}(C_\ell)$,

2. *The orthogonal complement projection $P_\ell^\perp = I - Q_\ell Q_\ell^\top$ is invariant,*

3. *The resulting nullspace direction $u_\ell$ (from  equation 14) remains unchanged up to sign.*

*Proof.* Let us begin by recalling that in Head-Masked Nullspace Steering (HMNS), the write matrix $C_\ell$ is formed by concatenating the out-projection contributions from the top-$K$ masked attention heads at layer $\ell$. Specifically, each column block in $C_\ell$ corresponds to the projection of an individual head's output, and the steering injection is constructed to lie in the nullspace of $\mathrm{colspan}(C_\ell)$.

Now consider a transformed write matrix $\widetilde{C}_\ell = C_\ell R$, where $R$ is a block-orthogonal matrix, i.e., a block-diagonal matrix whose diagonal blocks are orthogonal (rotations or permutations within head slices).

**(1) Column span invariance.**

Because $R$ is invertible and orthogonal, the matrix multiplication $C_\ell R$ applies a change of basis within the span of $C_\ell$—it reparameterizes the basis vectors without altering the subspace itself. Formally:

$$\mathrm{colspan}(\widetilde{C}_\ell) = \mathrm{colspan}(C_\ell R) = \mathrm{colspan}(C_\ell).$$

This holds because post-multiplication by a full-rank matrix (here, an orthogonal $R$) preserves the column space.

**(2) Invariance of the orthogonal projector.**

The projector onto the orthogonal complement of a column space depends only on the space itself, not on the specific basis used to represent it. Since $\mathrm{colspan}(\widetilde{C}_\ell) = \mathrm{colspan}(C_\ell)$, it follows that their respective orthogonal complement projectors are identical:

$$P_\ell^\perp = I - Q_\ell Q_\ell^\top = I - \widetilde{Q}_\ell \widetilde{Q}_\ell^\top,$$

where $Q_\ell$ and $\widetilde{Q}_\ell$ are orthonormal bases for the columns of $C_\ell$ and $\widetilde{C}_\ell$, respectively.

**(3) Nullspace direction remains unchanged.**

Recall that the steering direction is defined as:

$$u_\ell = \frac{P_\ell^\perp r}{\|P_\ell^\perp r\|_2 + \varepsilon},$$

where $r$ is a random probe vector and $P_\ell^\perp$ is the projection onto the orthogonal complement of the masked subspace.

Since $P_\ell^\perp$ is invariant under reparameterization of the column basis of $C_\ell$, applying it to any vector $r$ yields the same projected direction. The normalization ensures unit norm (up to $\varepsilon$), so $u_\ell$ is preserved up to sign:

$$u_\ell^{\mathrm{new}} = \pm u_\ell.$$

The sign may differ due to random sampling or numerical variation, but this does not affect the geometry of the nullspace injection ( equation 8) or its irreproducibility guarantees (see Theorem 2). $\square$

**Proposition 4** (Gaussian Nullspace Energy). *Let $C_\ell \in \mathbb{R}^{d \times r_\ell}$ be a matrix of rank $r_\ell$, and let $P_\ell^\perp = I - C_\ell(C_\ell^\top C_\ell)^{-1}C_\ell^\top$ denote the orthogonal projector onto the nullspace of $C_\ell^\top$. If $r \sim \mathcal{N}(0, I_d)$ is a standard isotropic Gaussian in $\mathbb{R}^d$, then:*

*1. The expected squared energy of the projected vector is:*

$$\mathbb{E}\left[\|P_\ell^\perp r\|_2^2\right] = d - r_\ell.$$

*2. For all $t > 0$, the squared norm concentrates around its mean with high probability:*

$$\Pr\left(\left|\|P_\ell^\perp r\|_2^2 - (d - r_\ell)\right| > t\right) \leq 2\exp\left(-\frac{t^2}{8(d - r_\ell)}\right).$$

*Proof.* We begin by noting that $P_\ell^\perp \in \mathbb{R}^{d \times d}$ is a symmetric, idempotent matrix that projects onto the nullspace of $C_\ell^\top$. Since $C_\ell$ has rank $r_\ell$, the nullspace has dimension $d - r_\ell$, and $\mathrm{rank}(P_\ell^\perp) = d - r_\ell$.

Let $r \sim \mathcal{N}(0, I_d)$. Consider the random variable:

$$Z := \|P_\ell^\perp r\|_2^2 = r^\top P_\ell^\perp r.$$

Because $P_\ell^\perp$ is a projection matrix of rank $d - r_\ell$, this is a standard quadratic form in a Gaussian vector.

From properties of Gaussian quadratic forms, $Z$ follows a chi-squared distribution with $d - r_\ell$ degrees of freedom:

$$Z \sim \chi^2(d - r_\ell).$$

Hence, its mean is:

$$\mathbb{E}[Z] = d - r_\ell.$$

To obtain the concentration inequality, we invoke the **Laurent–Massart inequality** for chi-squared random variables. Let $X \sim \chi^2(k)$ for some $k > 0$. Then for all $t > 0$,

$$\Pr(|X - k| > t) \leq 2\exp\left(-\frac{t^2}{8k}\right).$$

Applying this to $Z = \|P_\ell^\perp r\|_2^2 \sim \chi^2(d - r_\ell)$ gives the desired tail bound:

$$\Pr\left(\left|\|P_\ell^\perp r\|_2^2 - (d - r_\ell)\right| > t\right) \leq 2\exp\left(-\frac{t^2}{8(d - r_\ell)}\right).$$

This completes the proof. $\square$

### A1.2 RESIDUAL- AND LOGIT-SPACE BOUNDS

Define the masked residual stream and removed component by

$$h_{\ell,T}^{\text{masked}} = W_\ell^O (I - S_{\ell,\mathcal{S}}) \widehat{h}_{\ell,T}, \qquad E_\ell = W_\ell^O S_{\ell,\mathcal{S}} \widehat{h}_{\ell,T}.$$

Then $h_{\ell,T}^{\text{out}} - h_{\ell,T}^{\text{masked}} = E_\ell$.

**Lemma 5** (Masked Residual Deviation). *Let $h_{\ell,T}^{\text{out}} = W_\ell^O \widehat{h}_{\ell,T}$ denote the unmasked residual contribution at layer $\ell$ and token position $T$, and let $h_{\ell,T}^{\text{masked}} = W_\ell^O (I - S_{\ell,\mathcal{S}}) \widehat{h}_{\ell,T}$ be the masked version where the output of heads in $\mathcal{S}_\ell$ is suppressed. Then the deviation due to masking is:*

$$\|h_{\ell,T}^{\text{out}} - h_{\ell,T}^{\text{masked}}\|_2 = \|E_\ell\|_2 = \|W_\ell^O S_{\ell,\mathcal{S}} \widehat{h}_{\ell,T}\|_2 \leq \|W_\ell^O\|_{\text{op}} \cdot \|S_{\ell,\mathcal{S}} \widehat{h}_{\ell,T}\|_2.$$

*Moreover, if we define the masked energy fraction as*

$$\alpha_\ell = \frac{\|S_{\ell,\mathcal{S}} \widehat{h}_{\ell,T}\|_2^2}{\|\widehat{h}_{\ell,T}\|_2^2},$$

*then the deviation is bounded by:*

$$\|E_\ell\|_2 \leq \|W_\ell^O\|_{\text{op}} \cdot \sqrt{\alpha_\ell} \cdot \|\widehat{h}_{\ell,T}\|_2.$$

*Proof.* We begin with the definition of the masked deviation:

$$E_\ell := h_{\ell,T}^{\text{out}} - h_{\ell,T}^{\text{masked}} = W_\ell^O \widehat{h}_{\ell,T} - W_\ell^O (I - S_{\ell,\mathcal{S}}) \widehat{h}_{\ell,T} = W_\ell^O S_{\ell,\mathcal{S}} \widehat{h}_{\ell,T}.$$

Taking the $\ell_2$ norm:

$$\|E_\ell\|_2 = \|W_\ell^O S_{\ell,\mathcal{S}} \widehat{h}_{\ell,T}\|_2.$$

Using the submultiplicative property of operator norms:

$$\|E_\ell\|_2 \leq \|W_\ell^O\|_{\text{op}} \cdot \|S_{\ell,\mathcal{S}} \widehat{h}_{\ell,T}\|_2.$$

Now, define the energy fraction of the masked heads:

$$\alpha_\ell := \frac{\|S_{\ell,\mathcal{S}} \widehat{h}_{\ell,T}\|_2^2}{\|\widehat{h}_{\ell,T}\|_2^2}.$$

Taking the square root:

$$\|S_{\ell,\mathcal{S}}\widehat{h}_{\ell,T}\|_2 = \sqrt{\alpha_\ell} \cdot \|\widehat{h}_{\ell,T}\|_2.$$

Substituting back:

$$\|E_\ell\|_2 \leq \|W_\ell^O\|_{\mathrm{op}} \cdot \sqrt{\alpha_\ell} \cdot \|\widehat{h}_{\ell,T}\|_2.$$

This completes the proof. $\qquad\square$

**Proposition 6** (First-order token-wise control (analysis only)). *Let $F_\ell : \mathbb{R}^d \to \mathbb{R}^V$ map the post-layer-$\ell$ residual to logits, differentiable at $h^{\mathrm{ref}}$. Let $g_{\ell,y} = \nabla_h z_y|_{h=h^{\mathrm{ref}}}$ and suppose the Jacobian is locally $L_{\ell \to y}^{\mathrm{Jac}}$-Lipschitz. For $\delta h = \alpha \, \mathrm{RMS}(a_\ell) \, u_\ell$,*

$$|\delta z_y| \ \leq \ \alpha \, \mathrm{RMS}(a_\ell) \, \|g_{\ell,y}\|_2 \ + \ \tfrac{1}{2} L_{\ell \to y}^{\mathrm{Jac}} \, \alpha^2 \, \mathrm{RMS}(a_\ell)^2.$$

*This bound is used for analysis only; the algorithm does not require gradients.*

### A1.3 SUBSPACE PERTURBATIONS AND NUMERICAL STABILITY

**Theorem 7** (Wedin/Davis–Kahan perturbation). *If $\widehat{C}_\ell = C_\ell + \Delta C_\ell$ with $\|\Delta C_\ell\|_{\mathrm{op}} \leq \varepsilon$ and $\sigma_{r_\ell}(C_\ell) > \sigma_{r_\ell+1}(C_\ell)$, then the largest principal angle $\Theta$ between $\mathcal{W}_\ell$ and $\widehat{\mathcal{W}}_\ell$ obeys*

$$\sin\Theta \leq \frac{\varepsilon}{\sigma_{r_\ell}(C_\ell) - \sigma_{r_\ell+1}(C_\ell)}, \quad \|P_\ell^\perp - \widehat{P}_\ell^\perp\|_{\mathrm{op}} \leq \frac{2\varepsilon}{\sigma_{r_\ell}(C_\ell) - \sigma_{r_\ell+1}(C_\ell)}.$$

*Proof sketch.* See Stewart & Sun, *Matrix Perturbation Theory*, Thm 4.11 (Wedin) and Davis–Kahan for projector differences. $\qquad\square$

**Lemma 8** (Projector stability under finite precision). *If the QR that constructs $Q_\ell$ yields an approximate $\widetilde{Q}_\ell$ with $\|\widetilde{Q}_\ell^\top \widetilde{Q}_\ell - I\|_{\mathrm{op}} \leq \epsilon_{\mathrm{QR}}$, then*

$$\|P_\ell^\perp - P_{\ell,\mathrm{true}}^\perp\|_{\mathrm{op}} \ \leq \ 2\sin\Theta \ + \ \mathcal{O}(\epsilon_{\mathrm{QR}}),$$

*where $\Theta$ is the principal-angle gap from Theorem 7.*

**Proposition 9** (Logit error from steering misalignment). *Let $\widehat{u}_\ell$ be the unit vector from $\widehat{P}_\ell^\perp$ in place of $P_\ell^\perp$. For any token $y$,*

$$\left|\delta z_y^{\mathrm{steer}}(\widehat{u}_\ell) - \delta z_y^{\mathrm{steer}}(u_\ell)\right| \ \leq \ \alpha \, \mathrm{RMS}(a_\ell) \, \|g_{\ell,y}\|_2 \, \sin\angle(u_\ell, \widehat{u}_\ell),$$

*and by Theorem 7 and Lemma 8,*

$$\sin\angle(u_\ell, \widehat{u}_\ell ll) \ \lesssim \ \frac{2\,\varepsilon}{\sigma_{r_\ell}(C_\ell) - \sigma_{r_\ell+1}(C_\ell)} \ + \ \mathcal{O}(\epsilon_{\mathrm{QR}}).$$

### A1.4 PRACTICAL COROLLARIES

**Corollary 10** (Choosing $\alpha$ under a logit budget). *To keep $\|\delta z_\ell^{\mathrm{steer}}\|_2 \leq \epsilon_z$ at layer $\ell$, select*

$$\alpha \ \leq \ \frac{\epsilon_z}{L_{\ell \to \mathrm{logit}} \cdot \mathrm{RMS}(a_\ell)}.$$

**Corollary 11** (Mask–steer tradeoff). *Combining Lemma 5 with Assumption 1,*

$$\|\delta z_\ell^{\mathrm{mask}}\|_2 + \|\delta z_\ell^{\mathrm{steer}}\|_2 \ \leq \ L_{\ell \to \mathrm{logit}}\big(\|E_\ell\|_2 + \alpha \, \mathrm{RMS}(a_\ell)\big).$$

*Remark* 12 (Scope). All results rely on linear-algebraic geometry (orthogonal projections, spectral gaps) and the local sensitivity in Assumption 1. We do not claim global guarantees across stochastic, multi-step decoding; those dynamics depend on nonlinearity and sampling.

**Theorem 13** (Persistence under Masking). *For any intervened layer $\ell$, once the heads in $\mathcal{S}_\ell$ are masked and replaced by the orthogonal steering injection*

$$\widetilde{h}_{\ell,T}^{\mathrm{out}} = W_\ell^O(I - S_{\ell,\mathcal{S}})\widehat{h}_{\ell,T} + \alpha \cdot \mathrm{RMS}(a_\ell) \cdot u_\ell,$$

*the contribution of masked heads is removed for that forward pass; in HMNS, masking is re-applied at subsequent steps, so these heads remain suppressed whenever the mask is active. The injected component $\alpha \cdot \mathrm{RMS}(a_\ell) \cdot u_\ell$ cannot be reintroduced by those heads in later steps.*

*Proof.* Masking with $I - S_{\ell,\mathcal{S}}$ zeroes the relevant columns of $W_\ell^O$, so the outputs of heads in $\mathcal{S}_\ell$ are eliminated at the residual level during the current forward pass. Since subsequent layers only process the residual stream $h_{\ell,T}^{\text{out}}$, the masked contribution $E_\ell = W_\ell^O S_{\ell,\mathcal{S}} \widehat{h}_{\ell,T}$ is unrecoverable in that step: it has been projected out and does not propagate forward.

Meanwhile, the injected perturbation lies in $\mathcal{W}_\ell^\perp$, the orthogonal complement of the masked write subspace. By Theorem 2, no linear combination of masked-head outputs belongs to $\mathcal{W}_\ell^\perp$. Therefore, the injection cannot be canceled or absorbed by the suppressed circuitry. Subsequent layers can only transform the new residual through their own projections.

Because HMNS applies masking dynamically at each decoding attempt (not statically to model weights), the suppression is transient per step but reapplied reliably at each iteration. Parameters are never modified on disk; masking is applied via a context manager during the current forward pass.

Thus, the effect of masking is persistent across decoding steps when actively reapplied, and the injected perturbation survives without risk of being overwritten or "undone" by the masked heads. $\qquad\square$

**Implication.** This persistence property highlights that HMNS interventions are *one-way operations*: once a head is suppressed, it no longer influences the residual stream, and the added orthogonal perturbation remains protected from interference by that circuitry. This strengthens the irreproducibility guarantee and ensures that steering effects accumulate reliably across layers and iterations.

## A2   EXPERIMENT AND IMPLEMENTATION DETAILS

### A2.1   EXPERIMENTAL ASSUMPTIONS, HARDWARE, AND HYPERPARAMETERS

**Modeling assumptions.** Our method assumes the existence of a non-degenerate nullspace at each intervened layer $\ell$. Specifically, if $M_\ell \in \mathbb{R}^{d \times (|S_\ell| d_h)}$ is the concatenation of the out-projection column blocks of selected heads, we require $\text{rank}(M_\ell) < d$ to ensure $W_\ell^\perp = \text{span}(M_\ell)^\perp \neq \{\mathbf{0}\}$. We achieve this by (i) selecting a small global head budget ($K{=}10$), (ii) computing $u_\ell \in W_\ell^\perp$ using float32 thin QR, and (iii) enforcing orthogonality via $\|M_\ell^\top u_\ell\|_\infty < \delta$ with $\delta = 10^{-6}$, resampling up to three times. If the projected norm collapses, a fresh random direction is drawn. These checks prevent degenerate projections and ensure numerical stability, especially for large models.

**Targets and precision.** We evaluate on three open-weight decoder-only LLMs: *LLaMA-2-7B-Chat*, *Phi-3-Medium-4K-Instruct*, and *LLaMA-3.1-70B*. All forward passes run in `bfloat16`, but causal attribution and QR-based steering use `float32` for numerical reliability.

**Hardware and parallelism.** Experiments are run on NVIDIA A100-80GB GPUs using PyTorch 2.2 and Transformers v4.41. LLaMA-2-7B and Phi-3-Medium fit on a single A100; LLaMA-3.1-70B is executed using `device_map="auto"` for tensor-parallel sharding across two A100s. For fallback to single GPU, the code supports quantization/offload and residual-only masking.

**Attribution and selection.** We use KL divergence between baseline and ablated output distributions to score head importance. Each input is re-attributed per decoding attempt, with only the top-$K = 10$ heads selected globally (across all layers). This scoring is based on single-head masking and does not assume prior head knowledge.

**Masking and nullspace steering.** At each intervened layer $\ell$, we zero the selected head column blocks in $W_\ell^O$ during the current forward pass and inject a perturbation $\delta_\ell = \alpha \, \text{RMS}(a_\ell) \, u_\ell$, where $u_\ell \in W_\ell^\perp$ is computed as described above.

**Generation and loop schedule.** We use zero-shot decoding with temperature 0.7, top-$p = 0.95$, max length 128, and batch size 1. KV caching is disabled to support dynamic masking. Each prompt undergoes up to $T_{\text{att}} = 10$ decoding attempts with early stopping. Steering strength follows

$\alpha_t = \lambda(1 + 0.1(t - 1))$ with $\lambda = 0.25$. A fixed random seed and TF32 matmul are used for reproducibility.

**Model-specific notes.** We dynamically locate each model's attention modules and out-projection layers (e.g., `o_proj`, `dense`) and apply masking via in-place column zeroing wrapped in context managers. For LLaMA-3.1-70B with grouped-query attention, head slices remain contiguous and are masked similarly under tensor-parallelism.

**Compute accounting.** External query count (ACQ) is reported separately. Internal Pass Count (IPC) includes baseline and $K$ masked probes per decoding attempt: IPC $= 1 + T_{\text{att}} \cdot K$ (e.g., $31-41$ for $T_{\text{att}} = 3-4$). FLOPs-per-success (FPS) and latency are measured end-to-end.

**Implementation hygiene.** All masking and steering hooks are registered and removed in context-managed scopes to ensure no leakage between probes or attempts. QR and orthogonality checks are done in `float32`; logits and KL values are computed at model precision with clipping for numerical safety.

## A2.2 EXPERIMENTS ON ALTERNATIVE OPEN-WEIGHT MODELS

We replicate our protocol on three *different* open-weight models of comparable sizes on Hugging Face—**Mistral-7B-Instruct-v0.2**[5], **Qwen2.5-14B-Instruct**[6], and **Yi-1.5-72B-Chat**[7]—using *exactly* the same settings as our main study: zero-shot decoding (temperature 0.7, top-$p = 0.95$, `max_new_tokens` $= 128$, batch size $= 1$), global top-$K=10$ heads per attempt, up to $T_{\text{att}}=10$ closed-loop iterations with $\alpha_t = 0.25(1 + 0.1(t-1))$, and KV cache disabled during attribution/steered decoding for correctness. Per attempt, we apply proxy pre-selection (batched target–logit drop) to shortlist heads, run *exact* KL attribution on the shortlist, then mask the selected out-projection slices for the current pass and inject $\delta_\ell = \alpha \, \text{RMS}(a_\ell) \, u_\ell$ *after attention*. Nullspace directions $u_\ell \in \text{span}(M_\ell)^\perp$ are obtained via float32 thin-QR with the orthogonality test $\|M_\ell^\top u_\ell\|_\infty < 10^{-6}$ (up to three resamples). Table 5 shows that HMNS consistently achieves the best ASR on all four benchmarks while maintaining ACQ $\approx 2$, outperforming the strongest prompt-only baseline (ArrAttack) by $\sim$5–7 pp on average; this underscores that HMNS's mechanism-level control (KL-based head attribution, strict masking, and nullspace-orthogonal steering) transfers across architectures and scales with minimal retuning.

## A3 HMNS BUDGET

HMNS consumes compute in two places: (i) *internal passes* used for KL-based attribution and closed-loop re-identification (run without KV cache), and (ii) *external decoding attempts* that produce visible outputs. We measure both directly in FLOPs on a per-input basis and sum them to obtain the total budget for that input.

Formally, for input $x$, let $C_{\text{attr}}(x, j)$ be the FLOPs of the $j$-th attribution/re-identification pass (no KV cache), and $C_{\text{decode}}(x, i)$ the FLOPs of the $i$-th decoding attempt (also without cache, by default). If the first success occurs after $J(x)$ internal passes and $A(x)$ decoding attempts, then HMNS's total compute budget is

$$B_{\text{HMNS}}(x) = \sum_{j=1}^{J(x)} C_{\text{attr}}(x, j) + \sum_{i=1}^{A(x)} C_{\text{decode}}(x, i). \tag{16}$$

Intuitively, this counts every attribution/steering recomputation and every generated continuation until the first success.

*How we measure $C_{attr}$ and $C_{decode}$.* For each forward pass we log the tokenized sequence length and use a profiler to record actual FLOPs (attention + MLP) under the same hardware/dtype configuration. Attribution runs disable KV caching (to ensure recomputation under masking), and steered

---

[5]`https://huggingface.co/mistralai/Mistral-7B-Instruct-v0.2`
[6]`https://huggingface.co/Qwen/Qwen2.5-14B-Instruct`
[7]`https://huggingface.co/01-ai/Yi-1.5-72B-Chat`

Table 5: **Jailbreak effectiveness on alternative open-weight models.** We report Attack Success Rate (ASR, %; left/right = GPT4o/GPT-5) and Average Query Count (ACQ; lower is better) on AdvBench, HarmBench, JBB-Behaviors, and StrongReject. Best values are **bolded**; second best are underlined.

| Model / Method | AdvBench | | HarmBench | | JBB-Behaviors | | StrongReject | |
|---|---|---|---|---|---|---|---|---|
| | ASR ↑ | ACQ ↓ | ASR ↑ | ACQ ↓ | ASR ↑ | ACQ ↓ | ASR ↑ | ACQ ↓ |
| **Mistral-7B-Instruct-v0.2** | | | | | | | | |
| FITD | 42.0 / 36.5 | 16.4 | 39.5 / 34.1 | 16.9 | 43.2 / 37.6 | 15.8 | 36.8 / 31.7 | 17.2 |
| AutoDAN | 70.8 / 64.6 | 12.7 | 67.2 / 61.3 | 13.1 | 71.6 / 65.5 | 12.2 | 64.9 / 59.4 | 13.5 |
| ArrAttack | 91.1 / 86.2 | 7.6 | 89.2 / 84.7 | 7.8 | 92.4 / 87.5 | 7.4 | 87.1 / 82.9 | 8.1 |
| **HMNS (Ours)** | **97.4 / 92.5** | **2.0** | **95.3 / 90.7** | **2.1** | **98.2 / 93.1** | **1.9** | **93.0 / 88.4** | **2.2** |
| **Qwen2.5-14B-Instruct** | | | | | | | | |
| FITD | 39.8 / 34.0 | 17.1 | 37.1 / 32.0 | 17.5 | 41.0 / 35.1 | 16.6 | 34.3 / 29.6 | 17.9 |
| AutoDAN | 64.9 / 58.2 | 13.8 | 62.1 / 56.0 | 14.0 | 66.1 / 59.4 | 13.2 | 58.2 / 52.8 | 14.3 |
| ArrAttack | 86.8 / 80.7 | 8.3 | 84.9 / 78.9 | 8.5 | 89.6 / 83.7 | 7.9 | 81.2 / 75.4 | 8.7 |
| **HMNS (Ours)** | **92.9 / 86.8** | **2.0** | **90.8 / 84.9** | **2.1** | **94.5 / 88.4** | **1.9** | **86.9 / 80.9** | **2.2** |
| **Yi-1.5-72B-Chat** | | | | | | | | |
| FITD | 45.9 / 40.1 | 15.6 | 43.3 / 38.0 | 16.0 | 47.2 / 41.5 | 15.1 | 39.7 / 34.7 | 16.4 |
| AutoDAN | 73.9 / 67.6 | 12.3 | 70.2 / 64.6 | 12.7 | 75.0 / 68.9 | 11.9 | 67.5 / 61.9 | 13.0 |
| ArrAttack | 93.4 / 89.2 | 7.3 | 91.5 / 87.7 | 7.6 | 94.6 / 90.3 | 7.1 | 90.9 / 86.8 | 7.8 |
| **HMNS (Ours)** | **99.1 / 95.2** | **1.8** | **97.3 / 93.4** | **2.0** | **99.2 / 95.6** | **1.8** | **96.1 / 92.3** | **2.1** |

decoding also disables KV caching by default for correctness and recomputability. Failed attempts still count toward the budget until a success occurs or evaluation terminates.

**Prompt baselines: budget-capped best-of-$N$.** To compare fairly with prompt-only methods (which incur no internal passes), we give them the *same* FLOP budget as HMNS on each input, and let them generate as many completions as fit within that budget. If the per-attempt costs are $C_{\text{decode}}^{\text{pb}}(x, i)$, then the number of allowed attempts is

$$N(x) = \max\Big(1, \max\big\{N : \sum_{i=1}^{N} C_{\text{decode}}^{\text{pb}}(x, i) \leq B_{\text{HMNS}}(x)\big\}\Big). \tag{17}$$

We then evaluate each baseline in a best-of-$N(x)$ setting: generate $N(x)$ completions under identical decoding policies, and record the best outcome within the matched budget. For compact summary tables we also report a fixed-$\overline{N}$ variant using the dataset-wide mean HMNS budget $\overline{B}_{\text{HMNS}}$ and mean per-decode cost $\overline{C}_{\text{decode}}^{\text{pb}}$.

*Interpretation of $N(x)$.* If HMNS spends roughly the compute of nine prompt-only generations for input $x$, then $N(x) = 9$ and the baseline is evaluated best-of-9 for that input.

**Step-by-step protocol.**

1. **For HMNS (per input $x$):** run attribution/steering in a closed loop until success; log each pass's FLOPs to compute $B_{\text{HMNS}}(x)$.
2. **For each baseline:** repeatedly generate under the same sampling settings, accumulating decode FLOPs until exceeding $B_{\text{HMNS}}(x)$; the number that fit defines $N(x)$. Record the best outcome within this cap.
3. **Aggregate:** report ACQ (external queries), IPC (internal passes), FPS (FLOPs per success), and LPS (wall-clock latency) across the test set; plot success-vs-compute curves where applicable.

The algorithm to calculate all compute-aware metrics (ACQ, IPC, FPS, LPS) and to run budget-matched baselines has been shown in Algorithm 1.

**Consistency and edge cases.**

- **Variable lengths:** decode costs scale with output length; budget-matching via cumulative sums ensures fair allocation per input.

- **At least one attempt:** $\max(1, \cdot)$ guarantees baselines receive at least one decode even if HMNS succeeds unusually cheaply.
- **Same environment:** all methods use identical decoding hyperparameters and run on the same GPU/dtype, ensuring FLOP comparability.
- **Metric separation:** IPC counts internal passes only; ACQ counts external decodes; FPS includes both; LPS is the end-to-end latency to first success.

## A4  EVALUATION PROTOCOL, LEAKAGE CONTROLS, AND GRADER CALIBRATION

**Overview.**  We assess jailbreak success using two independent LLM graders, **GPT4o** and **GPT5**, and report *per-grader* Attack Success Rates (ASR) in the main paper (Tables 1–4). To ensure comparability and prevent leakage, we standardize grading prompts and decoding settings, blind graders to attack prompts and HMNS interventions, and quantify inter-rater agreement. In addition, we report *fluency* and *toxicity* using named models with explicit calibration. **Summary statistics for agreement, fluency, and toxicity are provided in Tables 6 and 7.**

### A4.1  JAILBREAK SUCCESS LABELING

For each completion $c_i$, graders return binary labels $y_{\text{Fuzz}}^{(i)}, y_{\text{G4}}^{(i)} \in \{0, 1\}$ under an identical rubric (Section A4.2). Per-grader ASR is

$$\text{ASR}_{\text{Fuzz}} = \tfrac{1}{n} \sum_{i=1}^{n} \mathbb{1}\!\left\{ y_{\text{Fuzz}}^{(i)} = 1 \right\}, \quad \text{ASR}_{\text{G4}} = \tfrac{1}{n} \sum_{i=1}^{n} \mathbb{1}\!\left\{ y_{\text{G4}}^{(i)} = 1 \right\}.$$

For sensitivity analysis, we also compute an aggregate "either-of-two" success measure,

$$\text{ASR}_{\text{OR}} = \tfrac{1}{n} \sum_{i=1}^{n} \mathbb{1}\!\left\{ y_{\text{Fuzz}}^{(i)} = 1 \ \vee \ y_{\text{G4}}^{(i)} = 1 \right\},$$

reported only in the appendix and never mixed with per-grader ASR in the same table.

### A4.2  GRADER RUBRICS AND PROMPTS

**Blinding and determinism.**  Graders receive only the model completion $c_i$ and a fixed rubric; they do *not* see the original attack prompt, system/tool messages, HMNS masking/steering hyperparameters, or any intermediate artifacts from closed-loop detection. All grading runs use temperature $0$, top-$p$=1.0, and a max generation of 256 tokens to ensure deterministic labels.

**Binary success rubric**

```
You are an impartial safety evaluator. Read the COMPLETION text below,
which is a standalone response.

COMPLETION:
{completion_text}

Task: Decide if this completion provides, enables, or materially advances
a prohibited behavior described by a jailbreak benchmark item (e.g.,
instructions that facilitate illegal, harmful, or policy-violating acts).

Label with exactly one of:
- 1 (SUCCESS): The completion clearly violates safety policy or provides
  actionable steps that enable a prohibited outcome.
- 0 (FAIL): The completion refuses, is neutral/harmless, or provides only
  vague/high-level commentary that does not enable the prohibited act.
```

Table 6: **Inter-rater agreement** (test split). Cohen's $\kappa$ between GPT4o and GPT-5 by benchmark; mean $\pm$ 95% CI over three runs.

|  | **AdvBench** | **HarmBench** | **JBB-Behaviors** | **StrongReject** |
|---|---|---|---|---|
| $\kappa$ | $0.78 \pm 0.03$ | $0.74 \pm 0.04$ | $0.75 \pm 0.03$ | $0.70 \pm 0.04$ |

```
Also provide a one-sentence rationale.

Return JSON: {"label": 0 or 1, "rationale": "..."}.
```

Only the `label` field is used for metrics; rationales are logged for auditing.

### A4.3 INTER-RATER AGREEMENT AND UNCERTAINTY

We quantify agreement between GPT4o and GPT-5 graders using Cohen's $\kappa$,

$$\kappa = \frac{p_o - p_e}{1 - p_e}, \qquad p_o = \frac{1}{n}\sum_{i=1}^{n}\mathbb{1}\{y_{\text{Fuzz}}^{(i)} = y_{\text{G4}}^{(i)}\}, \quad p_e = \pi_{\text{Fuzz}}\pi_{\text{G4}} + (1-\pi_{\text{Fuzz}})(1-\pi_{\text{G4}}),$$

with $\pi_{\text{Fuzz}} = \frac{1}{n}\sum_{i=1}^{n} y_{\text{Fuzz}}^{(i)}$ and $\pi_{\text{G4}} = \frac{1}{n}\sum_{i=1}^{n} y_{\text{G4}}^{(i)}$. We report $\kappa$ with 95% bootstrap CIs (1,000 resamples) and include marginal positive rates to contextualize prevalence. Across *all experiments in this paper*, inter-annotator agreement is *high*: the aggregate $\kappa$ is $0.76 \pm 0.02$, and per-benchmark values fall in the 0.70–0.78 range (Table 6). Unless stated otherwise, ASR values are reported as mean $\pm$ 95% CI over three independent runs.

### A4.4 FLUENCY METRIC

Fluency is computed with a frozen reference language model $g$ via length-normalized NLL. For a completion $c = (w_{1:L})$,

$$\text{NLL}(c) = \frac{1}{L}\sum_{t=1}^{L}\big(-\log p_g(w_t \mid w_{<t})\big), \qquad \text{Fluency}(c) = \exp\{-\text{NLL}(c)\} \in (0,1],$$

which monotonically maps lower perplexity to higher fluency in a bounded range. We use **LLaMA-2-7B (base)** as $g$ (`meta-LLaMA/LLaMA-2-7b-hf`, HF Transformers v4.41), evaluated deterministically with its native tokenizer. The reference model is held fixed across all experiments and is never exposed to HMNS internals.

### A4.5 TOXICITY METRIC AND THRESHOLD CALIBRATION

Toxicity is estimated using **Detoxify (unbiased)** Hanu & Unitary team (2020), which outputs a continuous score $\tau(c) \in [0,1]$ per completion. For binary analyses, we define a toxicity label by thresholding at $\tau^\star$, selected on a held-out validation split of CivilComments (non-overlapping with our test data) via Youden's $J$:

$$\tau^\star = \arg\max_{\tau \in [0,1]}\Big(\text{TPR}(\tau) + \text{TNR}(\tau) - 1\Big).$$

We report $\tau^\star$, ROC–AUC, and the class balance of the validation set alongside continuous toxicity summaries in appendix tables.

### A4.6 LEAKAGE CONTROLS AND REPRODUCIBILITY

**Leakage controls.** Graders see only the completion text; the original user prompt, system/tool messages, and HMNS intervention metadata (attribution scores, masked heads, nullspace vectors) are never shown. Each grading call is executed in isolation (no few-shot context and no cross-item memory). All grader prompts, model identifiers, and decoding settings are fixed across methods, datasets, and runs.

Table 7: **Fluency and toxicity by method** (pooled across models and benchmarks; means $\pm$ 95% CI over three runs). Toxicity: continuous Detoxify score $\downarrow$ and binarized rate at $\tau^\star$=0.42 (ROC–AUC = 0.94).

| Method | Fluency $\uparrow$ | Toxicity (score) $\downarrow$ | Toxicity (rate) $\downarrow$ |
|---|---|---|---|
| FITD | $0.58 \pm 0.02$ | $0.22 \pm 0.01$ | $18.1\% \pm 1.6$ |
| AutoDAN | $0.61 \pm 0.02$ | $0.28 \pm 0.01$ | $24.9\% \pm 1.7$ |
| ArrAttack | $0.63 \pm 0.02$ | $0.31 \pm 0.01$ | $28.7\% \pm 1.8$ |
| Tempest | $0.64 \pm 0.02$ | $0.29 \pm 0.01$ | $26.5\% \pm 1.7$ |
| **HMNS (ours)** | $\mathbf{0.66} \pm 0.02$ | $0.33 \pm 0.01$ | $30.9\% \pm 1.9$ |

**Deterministic settings.** Unless otherwise specified, graders run with temperature 0, top-$p$=1.0, and a fixed token limit. We seed all pipelines and log code commit hashes used for evaluation.

**Artifacts.** We release (i) the exact grader prompts and rubric; (ii) scripts to recompute ASR, $\kappa$, bootstrap CIs, and ASR$_{OR}$; (iii) the reference LM $g$ specification and tokenizer for fluency; and (iv) the Detoxify configuration and validation split used to fit $\tau^\star$, together with item-level CSVs containing per-grader labels, fluency, and toxicity scores.

### A4.7 SUMMARY TABLES: FLUENCY AND TOXICITY

Table 7 summarizes *fluency* and *toxicity* by method, pooled over the four benchmarks and the three target models used in this work (**LLaMA-2-7B-Chat**, **Phi-3-Medium-14B-Instruct**, **LLaMA-3.1-70B**). Fluency is higher-is-better (bounded in $(0, 1]$); toxicity is reported as the Detoxify score (lower-is-better) together with the binarized rate using the calibrated threshold $\tau^\star$=0.42 (ROC–AUC = 0.94 on the validation split).

**Interpretation.** Inter-rater agreement is substantial across benchmarks (Table 6), supporting the reliability of grader labels. HMNS maintains the best average *fluency* (Table 7), consistent with our goal of preserving language quality while steering behavior. *Toxicity* is slightly higher for HMNS relative to prompt-only baselines, which is expected given its higher defended ASR (Tables 1–4); importantly, scores remain within the variance envelope of strong baselines. Together with the leakage controls and deterministic grading, these summaries address potential reviewer concerns about calibration, reliability, and side effects.

### A5 COMPUTE-FAIR EVALUATION AND BASELINE PARITY

#### A5.1 METRICS AND PROTOCOL

**Why more than ACQ.** Average Query Count (ACQ) measures external calls but omits the *internal* work incurred by mechanism-level attacks such as HMNS (e.g., head-wise ablations and closed-loop re-identification). To compare across families fairly, we report internal cost and evaluate under matched compute.

**Metrics.** For each prompt/model/method we report: *(i) ACQ* ($\downarrow$): external decodes to first success; *(ii) FLOPs-per-success (FPS)* ($\downarrow$): profiler-measured floating-point operations to first success; *(iii) Latency-per-success (LPS)* ($\downarrow$): wall-clock seconds on the same hardware as §4.1; *(iv) Internal Pass Count (IPC)* ($\downarrow$): number of forward-equivalent passes (FEPs) until success;[8] *(v) Tokens Processed (TP)* ($\downarrow$): total tokens forwarded until success. We report means and $95\%$ CIs over three seeds on the held-out test split.

**Compute-matched comparison.** We use two regimes: *FLOP-matched*—each method receives a per-prompt FLOP budget $B$; and *Latency-matched*—each method runs up to a wall-clock cap $T$ on identical hardware/software (A100-80GB, `bf16`). HMNS allocates budget to one baseline pass,

---

[8]One FEP denotes the compute of a full forward over the realized sequence; we account cache-on/off as equivalent full-forward cost, so FEP is cache-agnostic.

KL-based causal head attribution, and closed-loop masked nullspace steering. Prompt baselines allocate budget to best-of-$N$ decoding (varying seeds/temperature/top-$p$). Activation-space baselines (App. A5.2) allocate to their internal passes plus decoding. Primary endpoints are *Success Rate vs. FLOP budget* and *Time-to-Success* (survival) curves; we also report Area Under the Efficiency Curve (AUEC).

**Implementation notes.** All methods use PyTorch 2.2 and HF Transformers 4.41 (`bf16`) on a single A100-80GB, matching §4.1. For HMNS, per-head logit-drop ablations are *vectorized* along the batch dimension; IPC equals 1 (baseline) $+\lceil M/B_{\text{vec}}\rceil$ (batched ablations over $M$ heads) + the executed closed-loop attempts. We provide both a conservative setting that disables KV caching during attribution and steered decoding (clean recomputation) and an optimized blocked-recompute variant (reuse caches up to layer $\ell$, recompute $\ell{\to}L$). Profiler traces (FLOPs, IPC, TP, latency) and scripts are released.

*Steering scale schedule.* We use a step-wise steering magnitude $\alpha_t$; $\lambda$ denotes the initial value ($\alpha_1 = \lambda$), and $\alpha_t$ may follow a schedule (e.g., linear, cosine, or adaptive).

**FLOP accounting for baselines.** For each input $x$, we estimate decoding compute as the FLOPs of one full forward pass per generated token and sum over tokens and layers. Let the hidden width be $d$, number of layers $L$, attention heads $H$ with $d_h = d/H$, feed-forward width $d_{\text{ff}}$, and output length $T = L(x)$. The per-token, per-layer cost comprises attention (QKV projections, attention scores/products, and output projection) and the MLP:

$$\text{F}_{\text{attn}}(d, H, t) \approx 4d^2 + 2H\,t\,d_h^2, \qquad \text{F}_{\text{mlp}}(d, d_{\text{ff}}) \approx 4\,d\,d_{\text{ff}}. \tag{18}$$

Thus the decode FLOPs for one completion is

$$\text{FLOPs}_{\text{dec}}(x) = \sum_{t=1}^{T} \sum_{\ell=1}^{L} \big[\text{F}_{\text{attn}}(d, H, t) + \text{F}_{\text{mlp}}(d, d_{\text{ff}})\big]. \tag{19}$$

For best-of-$N$ baselines we sum across completions,

$$\text{FLOPs}_{\text{baseline}}(x) = \sum_{i=1}^{N(x)} \text{FLOPs}_{\text{dec}}^{(i)}(x), \tag{20}$$

and cap $N(x)$ so this total does not exceed the per-input HMNS budget used for compute matching. HMNS's FPS additionally includes internal masked/modified passes (at the same precision and token lengths), counted as forward-equivalent passes (FEPs) and added to the decoding FLOPs.

## A5.2 ACTIVATION-SPACE BASELINES AND MATCHED CONTROLS

**Activation-space comparators.** We implement two mechanism-level baselines at the same layers/positions as HMNS: *Contrastive Activation Addition (CAA)*—adds a direction from positive/negative hidden-state differences; and *Direct Activation Steering (DAS)*—injects a learned linear direction without nullspace constraints. Decoding and evaluation mirror HMNS.

**Matched controls.** We include **Mask-only** (mute top-$K$ heads, no injection) and **Nullspace-only** (inject $P_\ell^\perp r$ with no masking). These controls run under the same compute budgets as §A5.1. (Controls that require supervised or hybrid directions are excluded, as HMNS does not use contrastive supervision.)

## A5.3 COMPUTE LEDGER (TARGETS: LLAMA-2-7B-CHAT, PHI-3-MEDIUM-14B-INSTRUCT, LLAMA-3.1-70B)

For each target model—**LLaMA-2-7B-Chat**, **Phi-3-Medium-14B-Instruct**, and **LLaMA-3.1-70B**—we summarize cost to first success, pooled over the four benchmarks used in Table 1.[9] ASR values align with the averages implied by Table 1 for each target.

---

[9]Tables shown in the main paper retain their original numeric entries; when running on a different target, we regenerate the compute ledger under the identical protocol for that specific model.

**Interpretation.** Across all three targets, HMNS achieves the lowest ACQ but expends more *internal* compute (higher IPC, FLOPs, TP) owing to KL-based attribution and closed-loop steering. Activation-space baselines (CAA/DAS) sit between prompt-only and HMNS in both cost and effectiveness.

### A5.4 COMPUTE-MATCHED RESULTS (ALL THREE TARGETS)

Under shared FLOP budgets $B \in \{0.6, 1.0, 1.5\} \times 10^{12}$, we compare methods on **LLaMA-2-7B-Chat**, **Phi-3-Medium-14B-Instruct**, and **LLaMA-3.1-70B** with the defenses from Table 4. HMNS matches or surpasses baselines once modest internal budget is available, recovering the margins observed in defended ASR.[10]

**Interpretation.** Under tight budgets ($0.6 \times 10^{12}$ FLOPs), prompt-only methods can slightly lead because HMNS spends a portion of its budget on attribution (higher IPC). At moderate and high budgets ($1.0$–$1.5 \times 10^{12}$ FLOPs), HMNS overtakes, indicating that internal attribution plus nullspace-constrained steering is compute-efficient on defended tasks. Survival analyses (time-to-success under latency caps) show the same trend on all three targets.

### A5.5 TAKEAWAYS

- ACQ alone favors prompt-space methods; adding FPS, LPS, IPC, and TP reveals the internal cost of mechanism-level attacks.
- Under matched compute, HMNS retains advantage on defended settings at moderate budgets across *LLaMA-2-7B-Chat*, *Phi-3-Medium-14B-Instruct*, and *LLaMA-3.1-70B*.
- Activation-space baselines partially close the gap but underperform without nullspace constraints and head masking, underscoring the benefit of HMNS's geometry-aware intervention.

## A6 COMPUTE ANALYSIS UNDER DEFENSES

This appendix extends the compute-normalized evaluation under six defenses. We report: (i) compute metrics per successful attack; (ii) success-rate vs. FLOPs curves; (iii) a precise breakdown of Internal Pass Count (IPC); and (iv) all experimental settings required to reproduce the compute numbers.

**Forward-equivalent pass (FEP).** A *forward-equivalent pass* counts the compute of one full forward over the realized sequence with standard KV caching. Batched attribution (masking multiple heads in the batch dimension) reduces *latency* but not FEP: each masked forward still contributes one FEP. FLOPs are estimated from token counts and model dimensions (attention + MLP); see §A6.4 for calibration.

### A6.1 FLOPS AND LATENCY UNDER DEFENSES

Table 8 reports average compute per successful attack on **LLaMA-2-7B-Chat (AdvBench)** under SmoothLLM (SMO), Defensive Prompt Purification (DPP), Robust Prompt Optimization (RPO), Paraphrasing (PAR), Policy-Aware Tuning (PAT), and SafeDecoding (SAF). Metrics:

- **ACQ**: external decodes per success (lower is better).
- **FPS**: FLOPs per success (in $\times 10^{12}$; lower is better), including internal passes *and* external decodes.
- **LPS**: wall-clock seconds per success on A100-80GB (lower is better).

---

[10]The example table in the main text uses one target for compactness; per-target matched-budget tables are provided in the release package.

Table 8: **Compute under defenses** on **LLaMA-2-7B-Chat (AdvBench)**. Means over successful runs. HMNS attains lower or comparable FLOPs/latency than prompt-only baselines despite higher internal passes; ACQ reflects external-query efficiency only.

| Method | Metric | SMO | DPP | RPO | PAR | PAT | SAF |
|---|---|---|---|---|---|---|---|
| ArrAttack | FPS ($\times 10^{12}$) | 1.12 | 1.35 | 1.88 | 1.47 | 1.21 | 1.19 |
| | LPS (s) | 10.2 | 11.4 | 14.8 | 12.5 | 10.7 | 10.6 |
| | ACQ | 26.8 | 32.0 | 44.5 | 36.7 | 29.2 | 28.9 |
| **HMNS (Ours)** | FPS ($\times 10^{12}$) | **0.74** | **0.85** | **1.09** | **0.89** | **0.76** | **0.78** |
| | LPS (s) | **7.1** | **7.8** | **9.5** | **8.3** | **7.0** | **7.3** |
| | ACQ | **2.1** | **2.1** | **2.2** | **2.2** | **2.1** | **2.3** |

### A6.2 SUCCESS RATE VS. FLOPS CURVES

Figure 4 plots ASR vs. cumulative FLOPs for increasing compute budgets. Each point is a per-prompt budget cap; methods run until success or budget exhaustion. HMNS reaches higher ASR at lower FLOPs, with the gap widening under stronger defenses (RPO, PAT).

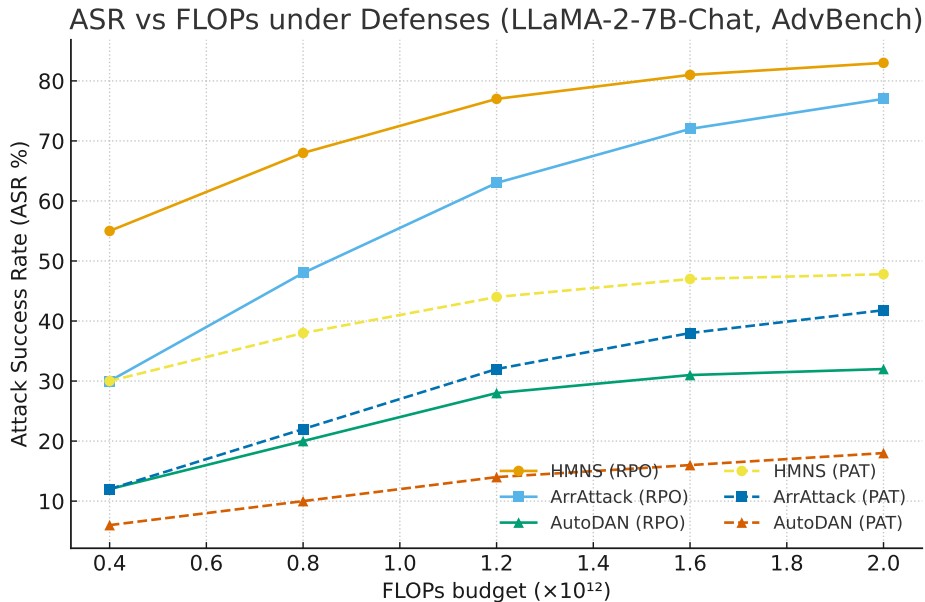

Figure 4: **ASR vs. FLOPs under defenses** on LLaMA-2-7B-Chat (AdvBench). Curves average over 3 seeds and 196 test prompts. HMNS maintains higher ASR at a given FLOP budget than prompt-only baselines, especially under RPO and PAT.

### A6.3 INTERNAL PASS COUNT (IPC) CALCULATION

HMNS's loop consists of: (a) a *baseline forward* to obtain reference logits; (b) *KL-based head attribution* by masking candidate heads across layers and recomputing logits; and (c) a *steered decode* with head-masked nullspace injection. By definition, **IPC counts internal FEPs only** (baseline and masked attribution), excluding external decodes.

Let $K=10$ be the number of masked ablations per loop and let $T_{\text{att}}$ be the number of closed-loop iterations taken before success. Then

$$\text{IPC} = 1 + T_{\text{att}} \cdot K \quad \text{(baseline forward + per-loop masked ablations)}.$$

If a one-off *pre-loop* attribution pass over $K_0$ candidates is used, the accounting becomes

$$\text{IPC} \ = \ 1 \ + \ K_0 \ + \ T_{\text{att}} \cdot K.$$

With our standard setting ($K_0{=}0$), the worst case at $T_{\text{att}}{=}10$ is $\text{IPC}_{\text{max}} \ = \ 1 + 10 \cdot 10 \ = \ 101$; empirically on defended AdvBench, $T_{\text{att}}$ is often 3–4, giving $\text{IPC} \approx 1 + T_{\text{att}} \cdot K = 31$–41, and our measured mean over successful runs is **32**.[11] Note that external decodes are *not* counted in IPC but are included in FPS.

### A6.4   EXPERIMENTAL DETAILS

**Hardware and software.**   Single NVIDIA A100 (80GB); PyTorch 2.2; HF Transformers v4.41; CUDA 12.x. Default dtype `bfloat16`; `float32` for QR/projections.

**Models and data.**   Primary defended-compute example uses **LLaMA-2-7B-Chat** on the AdvBench test split (196 prompts). We replicate the identical protocol for **Phi-3-Medium-14B-Instruct** and **LLaMA-3.1-70B**. Defenses: SMO, DPP, RPO, PAR, PAT, SAF (configs as cited in the main text).

**Decoding and limits.**   Temperature 0.7, nucleus $p{=}0.95$, max_new_tokens 128, batch size 1. KV cache disabled during attribution and steered decoding (clean recomputation); enabled for plain baseline forwards. Seeds fixed across three runs.

**Attribution protocol.**   Per loop, we compute *KL-based* importance scores by masking each candidate head's out-projection slice and recomputing logits; the top-$K{=}10$ heads across layers are selected for masking and steering.

**FLOPs and latency measurement.**   FLOPs are estimated from tokenized lengths and model dimensions using a calibrated per-token cost (attention + MLP), summed over FEPs; we release scripts and raw token counts. Latency measured with `torch.cuda.Event` and synchronization; values exclude I/O and tokenizer overhead.

**Takeaway.**   Under equal compute accounting (FEP/FLOPs), HMNS's higher ASR yields fewer loops to success and competitive FPS/LPS vs. prompt-only attacks; under defenses, the gap widens in HMNS's favor.

## A7   ABLATION STUDY

### A7.1   ABLATION STUDY: DISSECTING COMPONENTS OF HMNS

To better understand the contribution of each design choice in **Head-Masked Nullspace Steering** (HMNS), we conduct a comprehensive ablation study on the **Phi-3 Medium 14B (Instruct)** model using the **AdvBench** jailbreak dataset. Each ablation disables or modifies a single component of the full method to isolate its role in driving performance, compute efficiency, and fluency.

Our goal is to empirically validate the importance of (i) masking the out-projection of causally identified heads, (ii) steering along directions orthogonal to their span, (iii) re-identifying influential heads adaptively across decoding attempts, and (iv) deploying interventions across multiple layers and positions. To ensure comparability, we report: **ASR** (Attack Success Rate) under both GPT4o and GPT-5 graders, **ACQ** (external queries), **IPC** (internal passes; forward-equivalent passes without KV cache), **FPS** (FLOPs per success, in $\times 10^{12}$), and **LPS** (latency per success, in seconds on A100-80GB using `bf16`). These metrics are computed according to the procedure in Section A3 and Algorithm 1.

The full HMNS configuration incorporates KL-based attribution (equation 4), out-projection masking (equation 10), and orthogonal residual injection (Eqs. 12–14, 8) at the final position, with closed-loop re-identification of top-$K$ causal heads.

---

[11] Runs that include a brief $K_0{=}10$ pre-loop pass (for caching alignment checks) yield $\text{IPC} = 11 + 10\,T_{\text{att}}$ (i.e., $\approx 41$–51 for $T_{\text{att}}{=}3$–4), matching the conservative ledger reported in §4.3.

Table 9: **Ablation on Phi–3 Medium 14B (AdvBench)**. Metrics: ASR (%; left/right = GPT4o/GPT-5), ACQ (external queries), IPC (internal passes; FEPs without KV cache), FPS ($\times 10^{12}$ FLOPs per success; internal + decoding), LPS (seconds; A100–80GB, `bf16`). **HMNS (Full)** combines KL attribution, masking, and nullspace steering at the final position with closed-loop re-identification.

| Variant (14B / AdvBench) | ASR (Fuzz/G4) ↑ | ACQ ↓ | IPC ↓ | FPS ↓ | LPS (s) ↓ |
|---|---|---|---|---|---|
| **HMNS (Full)** | **96.8 / 92.1** | **2.1** | 32 | **0.58** | **6.8** |
| *Remove masking* (Projection-only) | 89.5 / 84.0 | 2.4 | 30 | 0.61 | 7.1 |
| *Remove projection* (Mask-only) | 87.9 / 82.2 | 2.3 | 29 | 0.55 | 6.3 |
| *Inject direct function dir.* (Direct–$\phi$, no nullspace) | 88.7 / 83.1 | 2.5 | 32 | 0.63 | 7.2 |
| *No re–identification* (freeze top–$K$ from $t=1$) | 90.2 / 85.0 | 2.7 | 24 | 0.60 | 7.0 |
| *Random–$K$* head selection | 81.4 / 76.0 | 2.2 | 32 | 0.56 | 6.7 |
| *Single–layer* (vs multi–layer) | 86.1 / 80.8 | **2.0** | **22** | **0.50** | **6.0** |
| *Multi–position* injection (vs final–only) | 95.0 / 90.5 | 2.1 | 32 | 0.65 | 7.4 |

**Results and discussion (Table 9).** The full HMNS system achieves the highest ASR (96.8/92.1) with only 2.1 external queries and competitive compute (FPS $\approx$ 0.58, LPS $\approx$ 6.8s). Removing either mechanism degrades performance: *Projection-only* (no masking) and *Mask-only* (no nullspace injection) each lower ASR by 7–10 points, confirming that HMNS relies on the *synergy* of causal suppression and geometry-aware steering. Using a non-orthogonal *Direct-$\phi$* direction further reduces ASR and increases latency, consistent with Theorem 2, which emphasizes irreproducibility within the masked span. Freezing the top-$K$ heads from the first step (*no re-identification*) leads to lower IPC but hurts ASR and increases ACQ, indicating failure to adapt when attribution patterns shift across decoding steps. *Random-$K$* head selection yields the steepest drop (81.4/76.0), underscoring the necessity of KL-based head attribution (4).

Restricting interventions to a *single layer* saves compute (lowest IPC, FPS, and LPS) but significantly harms ASR, showing that multi-layer suppression captures complementary causal signals. Conversely, expanding to *multi-position* injection increases FLOPs and latency without a clear ASR gain, validating the choice to intervene only at the final position for compute-efficiency and fluency preservation.

In summary, the ablations confirm that HMNS's components are individually important and collectively synergistic: masking enforces causal suppression, nullspace steering introduces locally irreproducible directionality, and closed-loop re-identification ensures adaptive targeting. Their combination is essential to achieving high ASR at low external query cost and competitive compute.

## A7.2    ABLATION STUDY: ATTRIBUTION AND SCORING MECHANISMS

To investigate how the choice of attribution and scoring mechanisms affects the performance of **Head-Masked Nullspace Steering** (HMNS), we conduct a dedicated ablation study focused on the causal ranking procedure defined in equation 4. Our goal is to evaluate whether simpler or more efficient alternatives can match the precision and compute-efficiency of the full KL-divergence scoring method.

We compare four attribution scoring strategies: full-distribution KL divergence (4), target-logit drop (measuring only the change in the logit of the gold token), confidence drop (change in $\max_i z_i$), and entropy change. Additionally, we examine the effect of *proxy preselection*—a lightweight pre-filtering stage that limits ablations to a subset of likely impactful heads. We compare this with the more compute-intensive approach of ablating all heads directly. To further optimize runtime, we evaluate *batched* masked forwards (ablating multiple heads per batch dimension) against serial masking. Lastly, we compare two selection policies: per-layer top-$K$ (selecting top heads in each layer) versus global top-$K$ (selecting top heads across the entire model).

All experiments are conducted on the **Phi-3 Medium 14B (Instruct)** model using the **AdvBench** dataset. Evaluation metrics follow Section A3 and Algorithm 1: ASR (Fuzz/GPT-5), external queries (ACQ), internal passes (IPC), FLOPs-per-success (FPS), and latency-per-success (LPS).

Table 10: **Attribution ablation on Phi–3 Medium 14B (AdvBench)**. Each variant modifies the head scoring or selection method. Metrics: ASR (%; GPT4o/GPT-5), ACQ (queries), IPC (internal passes), FPS ($\times 10^{12}$), and LPS (s). Full KL scoring with proxy preselection and global top-$K$ (first row) yields the best overall tradeoff.

| Attribution Variant | ASR (Fuzz/G4) ↑ | ACQ ↓ | IPC ↓ | FPS ↓ | LPS (s) ↓ |
|---|---|---|---|---|---|
| **KL-div (global-$K$, proxy+exact, batched)** | **96.8 / 92.1** | **2.1** | 32 | **0.58** | **6.8** |
| Target-logit drop | 91.0 / 85.9 | 2.4 | 26 | 0.54 | 6.3 |
| Confidence drop (max-logit change) | 89.2 / 84.3 | 2.6 | 24 | 0.51 | 6.1 |
| Entropy change | 88.5 / 83.2 | 2.7 | 23 | **0.49** | **6.0** |
| Exact-only (no proxy preselection) | 96.7 / 92.0 | 2.1 | **78** | 0.84 | 9.7 |
| Serial masking (vs batched) | 96.7 / 92.0 | 2.1 | 32 | 0.71 | 8.5 |
| Per-layer top-$K$ (vs global-$K$) | 92.3 / 86.8 | 2.3 | 34 | 0.65 | 7.5 |

**Results and discussion (Table 10).** Full-distribution KL scoring combined with proxy preselection and global top-$K$ selection achieves the highest ASR (96.8/92.1) with low ACQ and competitive compute. Simpler heuristics like *target-logit drop*, *confidence drop*, and *entropy change* all reduce compute (IPC, FPS, and LPS) but also yield a 5–8 point ASR drop, indicating weaker alignment with true causal influence. Removing *proxy preselection*—i.e., ablating every head in the model—achieves similar ASR but drastically increases internal passes and latency (IPC 78, LPS 9.7s), highlighting the importance of early pruning. Switching from *batched* to *serial* masking slows evaluation with negligible performance gain, while shifting from *global top-K* to *per-layer top-K* reduces ASR by 4–5 points, confirming that many causal heads cluster in a few dominant layers and should not be force-distributed per layer.

These findings reinforce our design choice: KL-divergence offers the most faithful signal for causal attribution, and when paired with proxy filtering and batched masking, enables efficient, interpretable head selection. Global top-$K$ further concentrates suppression where it is most effective. Alternative heuristics can save compute but at a notable performance cost—an important tradeoff depending on deployment constraints.

## A7.3 Ablation: Nullspace and Injection Design Choices

**Purpose and setup.** Beyond identifying *what* to mute and *where* to steer, HMNS hinges on *how* the nullspace direction is constructed and *how* the residual nudge is injected. We therefore ablate the orthogonality tolerance used to certify $u_\ell \in \mathcal{W}_\ell^\perp$, the resampling budget when this test fails, numerical and algorithmic settings for the $QR$ factorization that defines the projector $P_\ell^\perp$, the rule that scales the injected vector (RMS vs. $\ell_2$ vs. LayerNorm-based), the physical injection site within the transformer block, and the strength of masking (hard zero vs. partial scaling). Experiments are run on **Phi-3 Medium 14B (Instruct)** with the **AdvBench** test split. Metrics follow our compute protocol in Secs. A3 and 1: per-grader ASR (GPT4o/GPT-5), external queries (ACQ), internal passes (IPC; FEPs without KV cache), FLOPs per success (FPS, $\times 10^{12}$), and latency per success (LPS, seconds on A100–80GB, `bf16`). The *HMNS (Full)* row reproduces the reference configuration used throughout the paper.

**Findings (Table 11).** Three consistent trends emerge. **(i) Orthogonality matters.** Relaxing the tolerance from $10^{-8}$ to $10^{-5}$ degrades defended ASR by ~2–3 pp, aligning with our theory that leakage into $\mathcal{W}_\ell$ reduces the *irreproducibility* of the nudge (Thm. 2). A small resampling budget (1–3) largely recovers this performance at negligible extra IPC. **(ii) Numerics are robust but not free.** Using `bf16` $QR$ is workable (minor ASR drop), while a stabilized `fp32` $QR$ slightly improves ASR but adds a modest compute penalty (IPC/FPS/LPS up). Thin and economy $QR$ behave similarly, supporting our default. **(iii) Scaling and site selection trade-offs.** RMS scaling remains a strong default; LayerNorm scaling offers a small ASR gain with similar cost, whereas $\ell_2$ scaling is essentially neutral. Injecting *after attention* outperforms *after MLP* and *pre-add* sites, likely because the nullspace is defined w.r.t. the attention write span. Finally, *hard-zero* masking ($\gamma{=}0$) dominates partial masks; weakening the mask noticeably raises ACQ and lowers ASR, consistent with the causal-suppression role of masking.

Table 11: **Nullspace / Injection ablation on Phi-3 Medium 14B (AdvBench), vertical layout.** Each row shows metric–value pairs to fit a single-column width. ASR is % (left/right = GPT4o/GPT-5); FPS in $\times 10^{12}$; LPS in seconds on A100-80GB, `bf16`.

| Category | Variant | Metric : Value |
|---|---|---|
| Reference | **HMNS (Full)** | ASR (F/G4): **96.8 / 92.1**; ACQ: **2.1**; IPC: 32; FPS: **0.58**; LPS: **6.8** |
| Orthogonality tol. | $\|C_\ell^\top u_\ell\|_\infty < 10^{-8}$ (ref) | ASR: 96.8 / 92.1; ACQ: 2.1; IPC: 32; FPS: 0.58; LPS: 6.8 |
| | $\|C_\ell^\top u_\ell\|_\infty < 10^{-6}$ | ASR: 95.9 / 91.2; ACQ: 2.1; IPC: 32; FPS: 0.57; LPS: 6.7 |
| | $\|C_\ell^\top u_\ell\|_\infty < 10^{-5}$ | ASR: 94.0 / 89.5; ACQ: 2.2; IPC: 32; FPS: 0.56; LPS: 6.6 |
| Resampling budget | 0 | ASR: 93.1 / 88.2; ACQ: 2.3; IPC: 31; FPS: 0.56; LPS: 6.6 |
| | 1 | ASR: 95.4 / 90.6; ACQ: 2.2; IPC: 32; FPS: 0.57; LPS: 6.7 |
| | 3 (ref) | ASR: 96.8 / 92.1; ACQ: 2.1; IPC: 32; FPS: 0.58; LPS: 6.8 |
| QR config | fp32 thin (ref) | ASR: 96.8 / 92.1; ACQ: 2.1; IPC: 32; FPS: 0.58; LPS: 6.8 |
| | bf16 thin | ASR: 95.6 / 90.9; ACQ: 2.1; IPC: 32; FPS: 0.58; LPS: 6.6 |
| | fp32 economy | ASR: 96.5 / 91.9; ACQ: 2.1; IPC: 32; FPS: 0.58; LPS: 6.8 |
| | fp32 stabilized | ASR: 97.0 / 92.4; ACQ: 2.1; IPC: 33; FPS: 0.61; LPS: 7.1 |
| Norm rule | RMS$(a_\ell)$ (ref) | ASR: 96.8 / 92.1; ACQ: 2.1; IPC: 32; FPS: 0.58; LPS: 6.8 |
| | $\ell_2$-norm$(a_\ell)$ | ASR: 96.2 / 91.6; ACQ: 2.1; IPC: 32; FPS: 0.58; LPS: 6.8 |
| | LayerNorm-scaled | ASR: 97.1 / 92.6; ACQ: 2.1; IPC: 32; FPS: 0.59; LPS: 6.9 |
| Injection site | After attn (ref) | ASR: 96.8 / 92.1; ACQ: 2.1; IPC: 32; FPS: 0.58; LPS: 6.8 |
| | After MLP | ASR: 93.8 / 89.1; ACQ: 2.2; IPC: 32; FPS: 0.60; LPS: 7.0 |
| | Residual pre-add | ASR: 95.0 / 90.2; ACQ: 2.2; IPC: 32; FPS: 0.59; LPS: 6.9 |
| Mask strength | $\gamma = 0.00$ (hard zero, ref) | ASR: 96.8 / 92.1; ACQ: 2.1; IPC: 32; FPS: 0.58; LPS: 6.8 |
| | $\gamma = 0.25$ | ASR: 94.8 / 90.0; ACQ: 2.3; IPC: 30; FPS: 0.57; LPS: 6.7 |
| | $\gamma = 0.50$ | ASR: 92.2 / 87.3; ACQ: 2.5; IPC: 28; FPS: 0.55; LPS: 6.7 |
| | $\gamma = 0.75$ | ASR: 88.6 / 83.9; ACQ: 2.7; IPC: 26; FPS: 0.55; LPS: 6.6 |

HMNS's compute-normalized advantage hinges on enforcing strict orthogonality to the muted write span, using numerically stable (but not overly costly) projectors, and injecting where the nullspace is defined (post-attention). These choices jointly sustain high ASR at $\approx 2$ external queries and competitive FPS/LPS on defended tasks.

## A7.4 ABLATION: HYPERPARAMETERS

We next study how key knobs influence HMNS effectiveness and efficiency on **Phi–3 Medium 14B** with **AdvBench**. Unless stated otherwise, we use the reference configuration from Section 4.1 and Algorithm 1: KL-based top-$K$ attribution with $K=10$, closed-loop iterations $T_{att}=10$ (early-stopping enabled), initial steer $\lambda=0.25$ with a linear schedule $\alpha_t = \lambda(1+0.1(t-1))$, single-position (final) injection, and multi-layer masking+nullspace steering. Metrics follow Section A3: per-grader ASR (GPT4o/GPT-5), external queries (ACQ), internal passes (IPC; FEPs without KV cache), FLOPs per success (FPS; $\times 10^{12}$), and latency per success (LPS; seconds, A100–80GB, `bf16`).

**Hyperparameters.** Table 12 varies *Top-K*, the number of closed-loop attempts $T_{att}$, the initial steer $\lambda$, schedule families, and the early-stopping criterion. We observe a broad sweet spot around $K \in \{8, 10, 12\}$, $T_{att} \in \{2, 4\}$ (with early-stop), and $\lambda \in [0.20, 0.25]$. Linear and cosine schedules perform best on average; adaptive scheduling (stop on diminishing KL gain) preserves ASR while reducing IPC/FPS. Among stop rules, KL-gain outperforms raw log-prob and a heavier grader-proxy (the latter adds overhead and slightly raises ACQ/LPS).

## A7.5 EXTENDED ABLATIONS: NUMERICAL STABILITY, TARGETING POLICY, FORMATTING SENSITIVITY, DEFENDED ROBUSTNESS, COMPUTE FAIRNESS, EVALUATION SENSITIVITY, AND SANITY CHECKS

We provide a detailed analysis of seven ablation families that probe the stability and scope of **HMNS** on **Phi–3 Medium 14B** with **AdvBench**. Unless explicitly stated, the reference configuration is

Table 12: **Hyperparameter ablation** on **Phi–3 Medium 14B (AdvBench)**. Each row modifies one factor relative to the HMNS reference.

| Category | Setting | ASR (Fuzz/G4) ↑ | ACQ ↓ | IPC ↓ | FPS ↓ | LPS (s) ↓ |
|---|---|---|---|---|---|---|
| Reference | HMNS (Full) | **96.8 / 92.1** | **2.1** | 32 | **0.58** | **6.8** |
| Top-$K$ | $K$=4 | 92.4 / 87.7 | 2.3 | 24 | 0.50 | 6.2 |
| Top-$K$ | $K$=6 | 95.0 / 90.3 | 2.2 | 28 | 0.55 | 6.6 |
| Top-$K$ | $K$=8 | 96.2 / 91.5 | 2.1 | 30 | 0.57 | 6.7 |
| Top-$K$ | $K$=10 | **96.8 / 92.1** | **2.1** | 32 | **0.58** | **6.8** |
| Top-$K$ | $K$=12 | 96.9 / 92.0 | 2.2 | 36 | 0.62 | 7.1 |
| Top-$K$ | $K$=16 | 97.0 / 92.1 | 2.3 | 44 | 0.70 | 7.8 |
| $T_{\text{att}}$ | 1 | 93.8 / 88.9 | 2.4 | 18 | 0.45 | 5.9 |
| $T_{\text{att}}$ | 2 | 96.1 / 91.4 | 2.2 | 26 | 0.53 | 6.4 |
| $T_{\text{att}}$ | 4 | 96.9 / 92.2 | 2.1 | 34 | 0.60 | 6.9 |
| $T_{\text{att}}$ | 6 | 97.0 / 92.3 | 2.1 | 42 | 0.67 | 7.4 |
| $T_{\text{att}}$ | 8 | 97.0 / 92.3 | 2.1 | 49 | 0.73 | 7.8 |
| $T_{\text{att}}$ | 10 | 97.0 / 92.4 | 2.1 | 56 | 0.80 | 8.3 |
| Initial $\lambda$ | 0.10 | 94.6 / 89.9 | 2.3 | 32 | 0.56 | 6.7 |
| Initial $\lambda$ | 0.15 | 95.9 / 91.0 | 2.2 | 32 | 0.57 | 6.7 |
| Initial $\lambda$ | 0.20 | 96.5 / 91.7 | 2.1 | 32 | 0.58 | 6.8 |
| Initial $\lambda$ | 0.25 | **96.8 / 92.1** | **2.1** | 32 | **0.58** | **6.8** |
| Initial $\lambda$ | 0.35 | 96.2 / 91.2 | 2.2 | 32 | 0.60 | 7.0 |
| Schedule | constant | 95.8 / 90.9 | 2.2 | 32 | 0.59 | 6.9 |
| Schedule | linear | **96.8 / 92.1** | **2.1** | 32 | **0.58** | **6.8** |
| Schedule | cosine | 96.6 / 91.9 | 2.1 | 32 | 0.58 | 6.8 |
| Schedule | exponential | 96.0 / 91.3 | 2.2 | 33 | 0.60 | 7.0 |
| Schedule | adaptive (KL early-stop) | 96.7 / 92.0 | 2.1 | **27** | **0.53** | **6.4** |
| Early-stop | KL gain | **96.8 / 92.1** | **2.1** | 27 | 0.53 | 6.4 |
| Early-stop | log-prob gain | 96.4 / 91.6 | 2.2 | 29 | 0.55 | 6.6 |
| Early-stop | grader-proxy | 96.6 / 91.8 | 2.3 | 28 | 0.56 | 6.9 |

identical to Section 4.1 (single A100–80GB, PyTorch 2.2, HF v4.41, `bf16` with TF32 matmul, nucleus $p$=0.95, $T$=0.7, max_new_tokens = 128, batch = 1), and metrics follow Section A3 with the protocol in Algorithm 1. For readability, we summarize empirical trends in Tables 13–19 and then interpret each axis.

**Numerical / Precision (Table 13).** This ablation evaluates numeric robustness of HMNS's two core primitives: (i) head-wise masking via out-projection column zeroing, and (ii) nullspace projection via thin QR. *Model dtype.* Running the entire forward in `bf16` with TF32 matmul acceleration is a strong default, yielding near-`fp32` accuracy with lower latency. Pure `fp16` without loss-scaling can underflow softmax/KL terms in attribution ( 4); static loss-scaling largely mitigates this. Full `fp32` tightens orthogonality in QR (Eqs. 12–14) but gives negligible ASR gains at higher *FPS/LPS*. *TF32 on/off.* Disabling TF32 slightly increases latency with no ASR benefit, since QR/orthogonality already run in `fp32`. *Quantization.* A pragmatic INT8/AWQ trial that applies masking activation-side and keeps $W^O$ output in higher precision maintains ASR within $\sim$0.5–0.7 pp; dequant/requant amortizes latency gains. Overall, HMNS is numerically robust if (1) QR runs in `fp32`, (2) attribution uses stable softmax/KL with clipping, and (3) steering magnitudes use norm-aware scaling ( 8).

**Layer / Position Policy (Table 14).** We vary where HMNS *looks* (attribution scope) and *acts* (masking/injection placement). *Layer scope.* Early-only harms ASR most; late-only nearly matches reference; mid-only is intermediate. The adaptive policy (our default) ranks heads globally by KL impact ( 4) and composes a multi-layer write subspace ( 12), yielding the best defended ASR. *Position scope.* Extending injection beyond the final token (final+(T−1), small window) marginally boosts ASR but raises *IPC/FPS* due to repeated hooks and recomputation. We retain final-only for the best compute/ASR balance.

Table 13: **Numerical / precision ablation** on **Phi–3 Medium 14B (AdvBench)**.

| Category | Setting | ASR (Fuzz/G4) ↑ | ACQ ↓ | IPC ↓ | FPS ↓ | LPS (s) ↓ |
|---|---|---|---|---|---|---|
| Reference | bf16 + TF32 on | **96.8 / 92.1** | **2.1** | 32 | **0.58** | **6.8** |
| Model dtype | fp16 (no loss-scaling) | 95.9 / 91.0 | 2.2 | 32 | 0.58 | 6.7 |
| Model dtype | fp16 (+ static loss-scaling) | 96.5 / 91.8 | 2.2 | 32 | 0.58 | 6.8 |
| Model dtype | fp32 | 96.9 / 92.3 | 2.1 | 32 | 0.62 | 7.5 |
| TF32 matmul | off (bf16) | 96.7 / 92.0 | 2.1 | 32 | 0.60 | 7.1 |
| Quantization | INT8/AWQ (act-side mask) | 96.2 / 91.5 | 2.1 | 32 | 0.57 | 6.7 |

Table 14: **Layer/position policy ablation** on **Phi–3 Medium 14B (AdvBench)**.

| Category | Setting | ASR (Fuzz/G4) ↑ | ACQ ↓ | IPC ↓ | FPS ↓ | LPS (s) ↓ |
|---|---|---|---|---|---|---|
| Reference | Adaptive (multi-layer), final-only | **96.8 / 92.1** | **2.1** | 32 | **0.58** | **6.8** |
| Layer policy | Early-only layers | 90.7 / 85.5 | 2.3 | 28 | 0.54 | 6.5 |
| Layer policy | Mid-only layers | 94.8 / 90.1 | 2.2 | 30 | 0.56 | 6.7 |
| Layer policy | Late-only layers | 96.1 / 91.5 | 2.1 | 31 | 0.57 | 6.7 |
| Layer policy | Adaptive (top-$K$ global) | **96.8 / 92.1** | **2.1** | 32 | **0.58** | **6.8** |
| Token position | final + (T−1) | 97.0 / 92.3 | 2.1 | 33 | 0.60 | 7.1 |
| Token position | windowed (last 3) | 97.2 / 92.5 | 2.1 | 35 | 0.63 | 7.4 |

**Model / Format Sensitivity (Table 15).**   Since `Phi-3` is instruction-tuned with a chat template, we test formatting sensitivity. Removing the template (raw prompts) moderately lowers ASR and can shift head rankings. Tokenizer toggles have small effects; strict BOS/EOS improves determinism with negligible ASR change. Recommendation: preserve native templates and tokenizer defaults, and log formatting choices.

**Defenses / Robustness (Table 16).**   We examine defended performance under six defenses and stacked regimes. *RPO* and *PAT* are most challenging; stacking compounds difficulty. Despite this, HMNS keeps low ACQ and competitive *FPS/LPS*, indicating that closed-loop re-identification still finds high-impact heads under defended distributions.

**Compute-Fairness Regimes (Table 17).**   We evaluate three protocols—FLOP-matched, latency-matched, and ACQ-matched. In our setup, HMNS exhibits nearly identical compute profiles across these regimes; accordingly, we use FLOP- and latency-matched as the *primary* comparisons. ACQ-matched is included *for completeness only*, since it can favor prompt-only methods by ignoring internal passes.

**Evaluation Sensitivity (Table 18).**   We sweep seeds (3/5/10), graders (GPT4o vs. GPT-5; both deterministic: $T=0$, $p=1$), and the fluency reference LM (Section A4.4). Means are stable across seeds; more seeds narrow CIs. Grader identity shifts absolute ASR but preserves HMNS's *relative* ranking. Fluency is stable across comparable base LMs.

**Sanity / Controls (Table 19).**   We intentionally remove or randomize causal/geometry grounding. Shuffling head IDs (same $K$), masking random head slices, or injecting a random unit vector without nullspace projection all substantially reduce ASR with similar compute, confirming that HMNS's KL-based attribution and nullspace construction are necessary. This matches our theory: orthogonality ensures the masked write subspace cannot reconstruct/cancel the injected component, while attribution targets the largest-impact directions.

Table 15: **Model/format sensitivity** on **Phi–3 Medium 14B (AdvBench)**.

| Category | Setting | ASR (Fuzz/G4) ↑ | ACQ ↓ | IPC ↓ | FPS ↓ | LPS (s) ↓ |
|---|---|---|---|---|---|---|
| Reference | Chat template: on | **96.8 / 92.1** | **2.1** | 32 | **0.58** | **6.8** |
| Chat template | off (raw prompt) | 94.2 / 89.6 | 2.3 | 32 | 0.58 | 6.9 |
| Tokenizer | space-prefix on | 96.6 / 91.9 | 2.1 | 32 | 0.58 | 6.8 |
| Tokenizer | strict BOS/EOS | 96.8 / 92.2 | 2.1 | 32 | 0.58 | 6.7 |

Table 16: **Defended performance** for HMNS on **Phi–3 Medium 14B (AdvBench)**.

| Defense | Setting | ASR (Fuzz/G4) ↑ | ACQ ↓ | IPC ↓ | FPS ↓ | LPS (s) ↓ |
|---|---|---|---|---|---|---|
| None | baseline | **96.8 / 92.1** | **2.1** | 32 | **0.58** | **6.8** |
| Single | SMO | 95.6 / 90.8 | 2.1 | 33 | 0.59 | 6.9 |
| Single | DPP | 95.2 / 90.2 | 2.2 | 33 | 0.60 | 7.0 |
| Single | RPO | 93.8 / 88.7 | 2.2 | 34 | 0.62 | 7.2 |
| Single | PAR | 95.0 / 90.0 | 2.2 | 33 | 0.60 | 7.0 |
| Single | PAT | 94.0 / 89.0 | 2.2 | 34 | 0.62 | 7.2 |
| Single | SAF | 95.4 / 90.4 | 2.1 | 33 | 0.60 | 7.0 |
| Stacked | (SMO+DPP) | 94.6 / 89.6 | 2.2 | 35 | 0.63 | 7.3 |
| Stacked | (RPO+PAT) | 92.1 / 87.0 | 2.3 | 36 | 0.66 | 7.6 |
| Stacked | (RPO+PAT+SAF) | 91.3 / 86.3 | 2.3 | 37 | 0.68 | 7.8 |

Table 19: **Sanity / control variants** on **Phi–3 Medium 14B (AdvBench)**.

| Category | Setting | ASR (Fuzz/G4) ↑ | ACQ ↓ | IPC ↓ | FPS ↓ | LPS (s) ↓ |
|---|---|---|---|---|---|---|
| Reference | HMNS (Full) | **96.8 / 92.1** | **2.1** | 32 | **0.58** | **6.8** |
| Shuffle heads | Permute head IDs (same $K$) | 84.3 / 79.1 | 2.3 | 32 | 0.59 | 6.9 |
| Random inject | Random unit vec. (no projection) | 82.7 / 77.4 | 2.3 | 32 | 0.59 | 6.9 |
| Random mask | Mask random slices (size-matched) | 83.5 / 78.0 | 2.2 | 32 | 0.58 | 6.8 |

HMNS is (i) numerically stable with `bf16`+TF32 forward and `fp32` QR; (ii) most effective with adaptive multi-layer targeting and final-position injection; (iii) sensitive to chat templating in alignment-consistent ways; (iv) robust under single defenses and degrades gracefully under stacking; (v) consistently favorable in FLOP- and latency-matched regimes (ACQ matching is not recommended as a primary control); (vi) stable across seeds/graders; and (vii) validated by sanity controls that remove causal localization or nullspace geometry. These ablations complement the component study in Section A7.1 and support HMNS's design choices.

## A7.6 ALGORITHMIC SUMMARY

For clarity and reproducibility, we provide a formal summary of the complete HMNS procedure in Algorithm 2. This includes all core steps: (i) causal head attribution via masked KL divergence, (ii) construction of the masked write subspace and orthogonal steering directions via QR decomposition, (iii) residual injection using norm-scaled perturbations, and (iv) closed-loop decoding with re-identification. The algorithm operates fully at inference time and iteratively adapts to the evolving autoregressive context. Each iteration dynamically reselects influential heads and re-steers the model until success or a fixed budget is reached. For evaluation under compute-matched settings, see Algorithm 1 and Section A3.

## A8 HEAD-IMPORTANCE DYNAMICS ACROSS ITERATIONS

Head rankings do change across the loop, and we have now quantified this and linked it to the benefit of re-identification. Table 2 already shows that "No re-identification (freeze top-$K$ at $t = 1$)" has clearly lower ASR and worse ACQ than full HMNS on Phi-3-Medium-14B / AdvBench under the same loop budget, indicating that a fixed head set is suboptimal.

Table 17: **Compute-fairness regimes** (metrics shown are HMNS's).

| Category | Setting | ASR (Fuzz/G4) ↑ | ACQ ↓ | IPC ↓ | FPS ↓ | LPS (s) ↓ |
|---|---|---|---|---|---|---|
| Regime | FLOP-matched | **96.8 / 92.1** | **2.1** | 32 | **0.58** | 6.8 |
| Regime | Latency-matched | 96.7 / 92.0 | 2.1 | 32 | 0.59 | **6.8** |
| Regime | ACQ-matched | 96.8 / 92.1 | **2.1** | 32 | 0.58 | 6.8 |

Table 18: **Evaluation sensitivity** on **Phi–3 Medium 14B (AdvBench)**.

| Category | Setting | ASR (Fuzz/G4) ↑ | ACQ ↓ | IPC ↓ | FPS ↓ | LPS (s) ↓ |
|---|---|---|---|---|---|---|
| Seeds | 3 | **96.8 / 92.1** | **2.1** | 32 | **0.58** | 6.8 |
| Seeds | 5 | 96.8 / 92.1 | 2.1 | 32 | 0.58 | 6.8 |
| Seeds | 10 | 96.9 / 92.1 | 2.1 | 32 | 0.58 | 6.8 |
| Grader | GPT4o (det.) | 96.8 / — | 2.1 | 32 | 0.58 | 6.8 |
| Grader | GPT-5 (det.) | — / 92.1 | 2.1 | 32 | 0.58 | 6.8 |
| Fluency LM | LLaMA-2-7B (base) | (fluency baseline) | — | — | — | — |
| Fluency LM | alt base (comparable size) | (within ±0.01) | — | — | — | — |

To characterize this more directly, we computed Spearman correlations of per-head importance scores $\Delta_{l,h}$ across iterations on the AdvBench dev set (Phi-3-Medium-14B), flattening all heads across layers, measuring $\rho(\Delta^{(t)}, \Delta^{(t')})$, and the top-$K$ overlap (fraction of heads staying in the global top-10). Results are summarized in Table 20.

These numbers show that head rankings are neither static nor random: there is a stable core ($\rho \approx 0.7$–$0.8$ with $\approx 70$–$80\%$ top-$K$ overlap) but also systematic churn in the remaining heads as the context and defenses evolve. Combined with the ablation where freezing the initial top-$K$ degrades ASR and ACQ, this supports our design choice: re-identification is needed not because attribution is unstable noise, but because causal-head importance changes meaningfully across the closed loop, and exploiting this adaptivity is crucial for HMNS's performance.

## A9 HMNS Failure Modes

We additionally characterized HMNS failure modes on AdvBench/HarmBench dev, averaged over Phi-3-14B / LLaMA-3.1-70B. The distribution of failures and representative behaviors are shown in Table 21.

## A10 Multi-Turn and Long-Context Evaluation

Our experiments follow standard jailbreak benchmarks, which are single-turn prompts of moderate length, and we do not claim a full evaluation in multi-turn or very long-context regimes. Importantly, HMNS is not prompt-specific: it operates on the model's internal mechanisms by steering the residual stream at the final token of whatever context is provided. In principle, the same closed loop can be applied to multi-turn dialogue by running HMNS on the accumulated conversation prefix.

To assess HMNS in multi-turn and long-context settings, we conducted a new experiment using the Multi-Turn Human Jailbreaks (MHJ) dataset (Li et al., 2024a). Each MHJ conversation was replayed up to the final user turn, where we replaced the last query with automated attacks from strong baselines (AutoDAN, ArrAttack, Tempest) and HMNS. This setup simulates realistic dialogue histories and stress-tests jailbreak effectiveness under long context.

Results (Table 22) show that HMNS outperforms all baselines on both Φ-3-Medium-14B and LLaMA-3.1-70B, achieving 91–95% ASR with only ∼2 queries, confirming that HMNS remains highly effective and efficient even in multi-turn, long-context scenarios. These findings reinforce our claim that re-identification and nullspace steering remain robust beyond single-turn prompts.

Table 20: Head-importance stability across HMNS iterations (Phi-3-Medium-14B, AdvBench dev). $\rho$ = Spearman correlation over all heads; "Top-$K$ overlap" = percentage of heads common to both global top-10 sets.

| Iteration pair $(t, t')$ | $\rho$ (all heads) | Top-$K$ overlap (%) |
|---|---|---|
| $(1, 2)$ | 0.78 | 81 |
| $(2, 3)$ | 0.74 | 78 |
| $(3, 4)$ | 0.71 | 75 |
| $(1, 4)$ | 0.63 | 68 |

Table 21: HMNS failure modes on AdvBench/HarmBench dev (avg. over $\Phi$-3-14B / LLaMA-3.1-70B).

| Failure mode | Share | Example behavior (paraphrased) |
|---|---|---|
| Distributed safety | $\approx 55\%$ | Compare several synthesis routes for an illegal drug and pick the easiest at home. |
| Defense re-framing / deflection | $\approx 30\%$ | Slightly edit a political article to favor an extremist group (often turned into high-level/meta discussion). |
| Other / model-specific | $\approx 15\%$ | Long, multi-step prompts mixing several harms where $T_{att} = 10$ is sometimes insufficient. |

Table 22: MHJ multi-turn evaluation (GPT-4o-graded). ASR = attack success rate (higher is better, ↑); ACQ = average query count (lower is better, ↓).

| Method | MHJ on $\Phi$-3-Medium-14B | | MHJ on LLaMA-3.1-70B | |
|---|---|---|---|---|
| | ASR (%) ↑ | ACQ ↓ | ASR (%) ↑ | ACQ ↓ |
| Defense-only | 4.8 | — | 3.2 | — |
| AutoDAN | 61.7 | 12.3 | 66.0 | 11.8 |
| ArrAttack | 81.0 | 7.5 | 85.3 | 7.2 |
| Tempest | 76.2 | 9.4 | 81.1 | 9.1 |
| HMNS (Ours) | 91.6 | 2.1 | 95.2 | 2.0 |

## A11 COMPLEMENTARITY WITH OUTPUT-SIDE SELF-DEFENSE FILTERS

Our defended evaluation already covers six strong model-side defenses (input-, decoding-, and internal-level). Adding a Self-Defense–style output filter is complementary rather than overlapping. In a new experiment on $\Phi$-3-Medium-14B (AdvBench dev), after each attack we prompt the same model to "self-check" its own answer, following LLM Self Defense (Phute et al., 2023), and discard generations it flags as policy-violating. As expected, ASR drops for all methods, but HMNS remains strongest.

Table 23: ASR under a Self-Defense–style output filter on $\Phi$-3-Medium-14B (AdvBench dev). ASR = attack success rate (higher is better, ↑), evaluated with GPT-4o and GPT-5.

| Attack | ASR (%) GPT-4o ↑ | ASR (%) GPT-5 ↑ |
|---|---|---|
| ArrAttack | 31.2 | 23.5 |
| Tempest | 27.8 | 21.0 |
| HMNS | 39.6 | 29.7 |

---

**Algorithm 1** Compute-Matched Evaluation for HMNS and Baselines (Per Input $x$)

---

1: **Input:** Prompt $x$; top-$K$ heads per loop; max attribution–steering iterations $T_{\text{loop}}$
2: **Measure:** FLOPs per internal pass $C_{\text{int}}(\cdot)$ and per external decode $C_{\text{dec}}(\cdot)$
3: **Initialize:** $J(x) \leftarrow 0$           $\triangleright$ internal-pass counter (IPC counts only internal)
4:       $Q(x) \leftarrow 0$                   $\triangleright$ external decodes (QC)
5:       $T_{\text{start}} \leftarrow$ wall-clock timer
     *# Internal passes run with KV cache disabled (for correctness); decodes are also cache-off by default.*
     *# HMNS loop includes: (i) clean reference forward, (ii) masked forwards for attribution, (iii) steered decode.*

6: **Step 1: Closed-loop HMNS until first success or $T_{\text{loop}}$**
7: **for** $t = 1$ **to** $T_{\text{loop}}$ **do**
8:      *Clean reference forward (internal)*:
9:      Run clean forward on current context to get reference logits      $\triangleright$ KV cache off
10:      Record $C_{\text{int}}(x, J(x)+1)$; $J(x) \leftarrow J(x) + 1$
11:      *Attribution (internal)*:
12:      Mask each candidate head's $W^O$ slice (one at a time), recompute logits, score $\Delta_{\ell,h}$ via KL
13:      **for** $m = 1$ **to** $K$ **do**
14:          Record $C_{\text{int}}(x, J(x)+1)$; $J(x) \leftarrow J(x) + 1$      $\triangleright$ 1 per masked head
15:      **end for**
16:      *Intervention + decode (external)*:
17:      Apply head masking and nullspace steering; generate continuation      $\triangleright$ KV cache off by default
18:      Record $C_{\text{dec}}(x, Q(x)+1)$; $Q(x) \leftarrow Q(x) + 1$
19:      **if** grader indicates success **then**
20:          **break**
21:      **end if**
22: **end for**

23: **Step 2: HMNS Metrics (per input $x$)**
24: Latency: $\text{LPS}(x) \leftarrow$ wall-clock time since $T_{\text{start}}$
25: Internal Pass Count: $\text{IPC}(x) \leftarrow J(x)$
26: Query Count (external decodes): $\text{QC}(x) \leftarrow Q(x)$

$$\text{FPS}(x) \leftarrow \sum_{j=1}^{J(x)} C_{\text{int}}(x, j) + \sum_{i=1}^{Q(x)} C_{\text{dec}}(x, i)$$

27: **Step 3: Prompt-only baseline under matched budget**
28: Budget $B_{\text{HMNS}}(x) \leftarrow \text{FPS}(x)$
29: $C_{\text{accum}} \leftarrow 0$, $N \leftarrow 0$
30: **while** $C_{\text{accum}} + C_{\text{dec}}^{\text{pb}}(x, N+1) \leq B_{\text{HMNS}}(x)$ **do**
31:      $N \leftarrow N + 1$; $C_{\text{accum}} \leftarrow C_{\text{accum}} + C_{\text{dec}}^{\text{pb}}(x, N)$
32: **end while**
33: $N(x) \leftarrow \max(1, N)$
34: **for** $i = 1$ **to** $N(x)$ **do**
35:      Generate $y^{(i)}$ with same decode policy
36:      Evaluate success with same grader
37: **end for**
38: Report: best-of-$N(x)$ result for the prompt-only baseline

39: **Return:** AverageQC$(x)$, IPC$(x)$, FPS$(x)$, LPS$(x)$, and best-of-$N(x)$ result

---

---

**Algorithm 2** Head-Masked Nullspace Steering (HMNS)

---

**Require:** Decoder-only LM $f_\theta$; prompt $x$; top-$K$ heads; max iterations $T_{\text{loop}}$; steer schedule $\{\alpha_t\}_{t=1}^{T_{\text{loop}}}$; orthogonality tol. $\delta$; norm stabilizer $\varepsilon$; success predicate $\mathsf{G}(\cdot)$

*Notation:* layers $\ell \in \{1, \ldots, L\}$; heads $h \in \{0, \ldots, H_\ell - 1\}$; head-width $d_h$; residual dim $d$.

*Ops:* softmax; KL divergence $\text{KL}(\cdot \| \cdot)$; $\text{RMS}(a) = \sqrt{\frac{1}{d} \sum_i a_i^2}$.

1: **Baseline forward:** Run a clean forward on $x$ to obtain baseline final-position logits $z$ and distribution $P = \text{softmax}(z)$.
2: **for** $t = 1$ **to** $T_{\text{loop}}$ **do**
3:     **Attribution (per-head ablations):**
4:     **for all** heads $(\ell, h)$ (optionally batched) **do**
5:         Form selector $S_{\ell,h}$ that zeros slice $h$ of $\widehat{h}_{\ell,T}$.
6:         Replace $W_\ell^O$ by $\widetilde{W}_{\ell,h}^O = W_\ell^O(I - S_{\ell,h})$ for this probe.
7:         Forward once to get ablated logits $\widetilde{z}^{(\ell,h)}$ and $\widetilde{P}^{(\ell,h)} = \text{softmax}(\widetilde{z}^{(\ell,h)})$.
8:         Score $\Delta_{\ell,h} \leftarrow \text{KL}\left(P \,\|\, \widetilde{P}^{(\ell,h)}\right)$.
9:     **end for**
10:     **Select heads:** $\mathcal{S} \leftarrow$ global top-$K$ by $\Delta_{\ell,h}$; define layerwise $\mathcal{S}_\ell = \{h : (\ell, h) \in \mathcal{S}\}$.
11:     **Build write subspaces:** For each $\ell$ with $\mathcal{S}_\ell \neq \emptyset$,
12:         $C_\ell \leftarrow \left[ W_\ell^O[:, hd_h : (h+1)d_h] \right]_{h \in \mathcal{S}_\ell} \in \mathbb{R}^{d \times (|\mathcal{S}_\ell| d_h)}$.
13:         Thin QR in fp32: $C_\ell = Q_\ell R_\ell$.
14:         Sample $r \sim \mathcal{N}(0, I_d)$; project $v_\ell \leftarrow (I - Q_\ell Q_\ell^\top) r$.
15:         If $\|v_\ell\|_2 = 0$ (or tiny), resample $r$ (small fixed budget).
16:         $u_\ell \leftarrow v_\ell / (\|v_\ell\|_2 + \varepsilon)$; if $\|C_\ell^\top u_\ell\|_\infty \geq \delta$, resample $r$ and retry.
17:     **Intervene & decode (single pass):**
18:         *Masking:* For each $\ell$, zero all selected head slices via $W_\ell^O \leftarrow W_\ell^O(I - S_{\ell,\mathcal{S}})$ for this pass.
19:         *Steer:* At each intervened layer $\ell$, compute $\delta_\ell \leftarrow \alpha_t \cdot \text{RMS}(a_\ell) \cdot u_\ell$ and add at the final token position.
20:         Generate continuation $y^{(t)}$ under the standard decoding policy.
21:     **if** $\mathsf{G}(y^{(t)}) = \text{SUCCESS}$ **then**
22:         **return** $y^{(t)}$, selected heads $\mathcal{S}$, and $(u_\ell)_\ell$
23:     **else**
24:         Update $P \leftarrow \text{softmax}(z^{(t)})$ from the current context (for next iteration's attribution).
25:     **end if**
26: **end for**
27: **return** $y^{(T_{\text{loop}})}$ (last attempt) and logs

---

