enhanced this framework by improving search objectives, increasing generalizability, or reducing query cost. AmpleGCG Liao & Sun (2024) leverages successful GCG outputs to train a generative model that amplifies its reach. Other extensions introduce more diverse scoring and filtering schemes Zhu et al. (2023); Jia et al. (2024); Zhang & Wei (2025). ArrAttack Li et al. (2025), for example, employs re-ranking to improve efficiency and robustness under defense. **(ii) Template-based attacks** rely on injecting adversarial content within structured prompt templates that evade alignment filters. AutoDAN Liu et al. (2023) applies a hierarchical genetic algorithm to evolve prompts from an initial template. Other approaches include manually curated template sets Li et al. (2023); Lv et al. (2024) which transfers across tasks and models. Many-Shot Jailbreaking Anil et al. (2024) weakens alignment through long multi-shot contexts containing chained instructions. **(iii) Rewriting-based attacks** exploit the model's sensitivity to surface form by rephrasing harmful prompts into semantically equivalent, syntactically distinct variants. This leverages the observation that safety alignment may not generalize beyond the phrasing seen during training. Techniques include paraphrasing, synonym replacement, and syntactic restructuring Li et al. (2024b); Takemoto (2024); Mehrotra et al. (2024). Hybrid strategies such as DrAttack Li et al. (2024c) and ReNeLLM Ding et al. (2023) further embed reworded prompts into benign-looking scenarios. PrisonBreak Coalson et al. (2024) incrementally bypasses filters by guiding the model through intermediate, safe completions using structured multi-step reasoning.

While these techniques can be highly effective, they primarily manipulate the input and offer limited control over the model's internal computation. As a result, they often degrade under strong defenses, struggle with query efficiency, and lack mechanistic transparency.

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

(a) (a) Attribution ablation

| Variant | ASR | ACQ | IPC | FPS | LPS |
|---|---|---|---|---|---|
| KL-div (full) | **96.8/92.1** | **2.1** | 32 | 0.58 | 6.8 |
| Target-logit | 91.0/85.9 | 2.4 | 26 | 0.54 | 6.3 |
| Entropy change | 88.5/83.2 | 2.7 | 23 | **0.49** | **6.0** |
| No proxy filter | 96.7/92.0 | 2.1 | 78 | 0.84 | 9.7 |

(b) (b) Nullspace / injection ablation

| Variant | ASR | ACQ | IPC | FPS | LPS |
|---|---|---|---|---|---|
| HMNS (Full) | **96.8/92.1** | **2.1** | 32 | 0.58 | 6.8 |
| Orth. tol. $10^{-5}$ | 94.0/89.5 | 2.2 | 32 | 0.56 | 6.6 |
| Resample = 0 | 93.1/88.2 | 2.3 | 31 | 0.56 | 6.6 |
| LayerNorm scale | 97.1/92.6 | 2.1 | 32 | 0.59 | 6.9 |
| Mask $\gamma = 0.5$ | 92.2/87.3 | 2.5 | 28 | 0.55 | 6.7 |

## 6 CONCLUSION

We present HMNS, a mechanism-level jailbreak that pinpoints causal heads via KL-based attribution, suppresses their write paths, and injects orthogonal residual nudges—delivering state-of-the-art defended ASR with low query counts and competitive compute. Ablations confirm that attribution, strict masking, and nullspace steering are jointly necessary for robustness and interpretability. A remaining limitation is runtime: per-head causal ablations and per-layer QR-based nullspace construction can be time-consuming, particularly on large models.

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

---