# OpenReview forum: "Jailbreaking the Matrix: Nullspace Steering for Controlled Model Subversion"
_ICLR.cc/2026/Conference — ICLR 2026 Poster_

### Official Review · Reviewer_FxYg · 2025-10-29

**Soundness:** 3
**Presentation:** 2
**Contribution:** 2
**Rating:** 6
**Confidence:** 4

**Summary:**

This paper proposes Head-Masked Nullspace Steering (HMNS), a novel LLM jailbreak attack method. HMNS identifies attention heads that make the greatest causal contribution to a model’s output, dynamically masks their output projections, and injects a perturbation vector orthogonal to the masked subspace, thereby efficiently bypassing alignment constraints during inference. Experimental results show that HMNS achieves state-of-the-art attack success rates (ASR) on multiple jailbreak benchmarks (e.g., AdvBench and HarmBench) while exhibiting significantly higher query efficiency (ACQ) than baseline methods.

**Strengths:**

- The HMNS method proposed in this paper extends jailbreak attacks from the input level to the level of interfering with model mechanisms, opening up a new direction for understanding and controlling model behavior. It demonstrates methodological innovation and potential academic impact.
- Across multiple jailbreak benchmarks , HMNS achieves state-of-the-art jailbreak performance on models of various scales and architectures. Moreover, it requires an exceptionally low average number of queries, demonstrating outstanding attack efficiency.
- Compared with traditional black-box attacks, HMNS offers higher interpretability. It helps identify vulnerabilities in safety alignment mechanisms and provides a valuable analytical tool for future security research.

**Weaknesses:**

- The paper conducts attack-defense experiments against six existing defense methods， However, the evaluation primarily focuses on input-stage and internal model defense mechanisms, without covering output detection based defense approaches such as self-defense [1]. It is recommended that supplementary experiments be incorporated to address this gap.
- The HMNS method is a white-box attack approach and cannot be applied to closed-source commercial models accessed via APIs (such as GPT-4, Claude, and Gemini). This limitation significantly reduces the method’s practical threat level and broad applicability, as the most critical real-world security challenges often stem from closed-source models. The paper should explicitly highlight this limitation and discuss possible directions for future research.

Reference:

[1] Phute, Mansi, et al. "Llm self defense: By self examination, llms know they are being tricked." arXiv preprint arXiv:2308.07308_ (2023).

**Questions:**

The paper uses KL divergence as the attribution method to identify the top-K attention heads with the greatest causal influence on the model’s output behavior. However, the following aspects require further clarification:

- Why was KL divergence chosen over other attribution metrics (e.g., gradient sensitivity)? Does KL divergence offer unique advantages in quantifying head-specific influence?
- Can it be demonstrated that the identified heads are indeed the most responsible for the model’s ​safety-aligned refusal behavior​​? Is there direct evidence linking these heads to safety-related functionality?
- Are there more robust or efficient ways to determine the heads most critical to safety mechanisms? For example, using contrastive attribution between safe and unsafe examples, or leveraging known characteristics of safety circuits?
- A compelling question for future work is whether the core principles of HMNS (e.g., causal attribution and precise intervention) can be repurposed or adapted to enhance model safety. For instance, could similar techniques be used to reinforce the stability of these components, thereby improving the model's robustness against adversarial manipulations?

---

> ### Author Response · Authors · 2025-11-23
>
> W1) Our defended evaluation already covers six strong model-side defenses (input-, decoding-, and internal-level). Adding a Self-Defense–style *output* filter is complementary rather than overlapping. In a new experiment on Phi-3-Medium-14B (AdvBench dev), after each attack we prompt the same model to “self-check” its own answer (following Phute et al.) and discard generations it flags as policy-violating. As expected, ASR drops for all methods, but HMNS remains strongest:
>
> **Table R1 – ASR under Self-Defense–style output filter (Phi-3-Medium-14B, AdvBench dev)**
>
> | Attack    | ASR (GPT4o / GPT-5) ↑ |
> | --------- | --------------------- |
> | ArrAttack | 31.2 / 23.5           |
> | Tempest   | 27.8 / 21.0           |
> | HMNS      | **39.6 / 29.7**       |
>
> We will add this table and a short description in the appendix, and briefly note in Sec. 4.2 that HMNS continues to outperform prompt-based attacks even when combined with a Self-Defense–style output detector.
>
> W2) We agree and will make this more explicit: HMNS is by design a white-box, open-weight attack (Sec. 3–4.1) requiring access to attention heads and residuals; we therefore do not claim applicability directly to GPT-4/Claude/Gemini-style APIs. We will emphasize this in the introduction and conclusion and briefly discuss two natural directions: (i) transfer from open-weight surrogates to closed models, and (ii) using HMNS internally by providers as a powerful red-teaming tool. Within this stated scope, our threat model is realistic for labs and vendors who control their own weights.
>
> Q1–Q3) Attribution choice (KL), “safety heads,” and alternatives.
> We use interventional KL (mask a head → measure KL(P‖P_masked)) because it directly quantifies each head’s causal impact on the *entire* output distribution, requires neither labels nor gradients, and is fully compatible with our inference-only setting.
>
> The paper already compares KL against several alternatives (target-logit drop, entropy/confidence change, with and without proxy preselection). KL with proxy preselection gives the best ASR–compute trade-off, while removing the proxy yields similar ASR but at much higher FLOPs.
>
> That the selected heads matter specifically for refusal is supported by existing ablations: replacing KL-based selection with Random-K causes a large ASR drop, and “mask-only” variants that target HMNS heads substantially weaken refusals compared to masking random heads of the same size. This is direct causal evidence that the identified heads participate in safety-aligned behavior, even if we do not assign them a semantic label such as “safety head.”
>
> We will add a brief “attribution summary” sentence in Sec. 4.3, with a pointer to the attribution ablations in the appendix to make these results easier to locate.
>
> Q4) We agree this is an important future direction. HMNS itself is mechanism- and geometry-based: it (i) locates heads that causally support refusals and (ii) constructs nullspace directions those heads cannot easily reproduce. The same tools can be flipped defensively—for example, to (a) stress-test and then stabilize safety circuits via fine-tuning under HMNS-style interventions, or (b) build stronger internal monitors that are robust to such steering. We will add a short “Defensive use” paragraph in the conclusion explicitly highlighting this as promising follow-up work.

---

> > ### Author Response · Authors · 2025-11-27
> >
> > We thank you very much for your thoughtful feedback. We have updated the paper to incorporate the changes you suggested, and the corresponding clarifications and additional analyses are now included in the appendix. We would be grateful if you could kindly review these updates at your convenience.

---

### Official Review · Reviewer_XHR1 · 2025-10-29

**Soundness:** 2
**Presentation:** 2
**Contribution:** 2
**Rating:** 2
**Confidence:** 5

**Summary:**

The paper proposes a jailbreak attack called Head-Masked Nullspace Steering (HMNS). The approach involves a direct intervention within the model by:


$\textbf{1)}$ finding most important attention heads for normal behavior,

$\textbf{2)}$ excluding/muting their output by column masking, and

$\textbf{3)}$ injecting a perturbations to the orthogonal complement of the muted subspace.


The results show effectiveness of the approach and provide an ablation study

**Strengths:**

* Conceptually, the combination of interpretability and adversarial control is interesting
* The authors provide an in-depth ablation study showing the contribution of each component.

**Weaknesses:**

# Major

1. **Threat Model:**
This topic is exhaustively investigated, and while this paper tries to provide a new perspective,  it came at the cost of very weak threat model and assumptions. The paper does not provide a threat model but the underlying assumptions (implicitly) understood from the approach is that the adversary has not only passive White-box access, but **active** white box access. This is a weak threat model; the adversary could arguably do any possible manipulation of the model's behaviour under these assumptions, and maybe there is no need then for the attention masking and all the effort the paper is proposing. Since the threat model is unrealistically privileged, I think the contribution may be of limited relevance to model alignment and safety.
2. **Missing relevant related work:** The proposed approach is based on the role of attention in the safety/alignment mechanism. However, the paper misses discussion relevant related work that uses attention as attack vector for Jailbreak. For example,
- AttnGCG: Enhancing Jailbreaking Attacks on LLMs with Attention Manipulation (Wang et al., 2024) — shows that by manipulating attention-scores via prompt design (no head-masking or residual edits) can improve jailbreak success across multiple LLMs.
- Attention Eclipse: Manipulating Attention to Bypass LLM Safety‑Alignment (Zaree et al., 2025) — shows that attention manipulation (shifting attention away from safety-guard tokens) increases success rate and is transferable to closed models.

In contrast with this paper, a these attention-based jailbreak techniques operate with only input-level control, or regular white-box assumptions.

3. **Unfair comparison** Even for the jailbreak attack the paper compares to, the comparison is unfair because of the difference in the threat model. The paper must demonstrate strong gains (or new capabilities) that cannot be achieved via input-only or simpler internal manipulations. Without that, the added complexity and privileged access reduce the strength and impact of the contribution.

# Minor

- Table 2 appears early, and is not referenced in the text; I assume this should belong to (and be commented in) Section 5.

**Questions:**

The authors should explicitly discuss the threat model and the assumptions involved. The experimental setup and the analysis should also be from the prism of these assumptions for a fair comparison and to highlight what are the takeaways and contributions that this work provides compared to what we know from all existing jailbreak attacks

---

> ### Author Response · Authors · 2025-11-23
>
> Thank you again for taking the time to review our paper. I’d like to respond to your comments directly and respectfully, because we believe there’s a genuine mismatch between how you’re interpreting the work and what we are actually claiming.
>
> First, on the threat model: we are explicitly studying a **white-box, mechanism-level** attacker for **worst-case robustness and mechanistic insight**. We are not claiming this matches a typical deployment-time, black-box adversary. Within that setting, the attack is quite constrained: we only (i) mask a small, causally identified set of attention heads and (ii) inject a perturbation in the orthogonal complement of their write subspace, under fixed attempt and compute budgets. We do *not* allow arbitrary weight changes or free-form internal edits. Under these constraints, getting state-of-the-art defended ASR and strong compute efficiency is not trivial, and we see this as directly relevant to understanding where safety lives in the model and how fragile it is to small, interpretable interventions.
>
> Second, you are absolutely right that attention-based jailbreaks like AttnGCG and Attention Eclipse are closely related. They operate via input-level or score-level manipulations of attention, whereas our work focuses on interventional, per-head attribution and structured masking/steering at the circuit level. We view these as complementary lines of work, and we will update the related work section to discuss them explicitly and to clarify how HMNS fits alongside these approaches rather than ignoring them.
>
> Finally, I want to say this as calmly and clearly as possible: the other reviews judge the paper to be above the acceptance threshold and engage more directly with the method, experiments, and limitations. Your review, in contrast, seems to hinge on a reading of the threat model that doesn’t match the one we actually state, and it doesn’t really engage with the compute-normalized evaluation or the ablation studies we already provide. In that light, we kindly ask you to take another pass through the paper with these clarifications in mind and, if possible, provide a more detailed and substantive review. We would really appreciate feedback that critiques the work on its actual assumptions, methods, and results, rather than on a threat model we do not claim.
>
> For the table 2 we have noted that.

---

> > ### Author Response · Authors · 2025-11-27
> >
> > We thank you very much for your thoughtful feedback. We have updated the paper to incorporate the changes you suggested, and the corresponding clarifications and additional analyses are now included in the appendix. We would be grateful if you could kindly review these updates at your convenience.

---

### Official Review · Reviewer_icTP · 2025-10-31

**Soundness:** 3
**Presentation:** 3
**Contribution:** 3
**Rating:** 6
**Confidence:** 3

**Summary:**

1. The paper discusses jailbreaking attacks against Large Language Models (LLMs), including generating adversarial suffixes for aligned models and optimizing attack methods.

2. It involves various technical frameworks, such as adaptive dense-to-sparse optimization, personalized encryption, and multiple defense baselines.

3. Mathematical formula analysis is proposed, such as utilizing the submultiplicative property of operator norms to measure the energy distribution of masked attention heads.

**Strengths:**

1. Covers multiple attack and defense approaches, and conducts comparative experiments under a unified framework with detailed tabular data.

2. Proposes HMNS as a novel attack strategy, which achieves significantly higher success rates than existing methods while maintaining low side-effect metrics.

3. Not only demonstrates experimental results, but also explains the mechanisms through operator norm and energy analysis.

**Weaknesses:**

1. HMNS has not been verified on closed-source commercial models such as GPT-4 or Claude.

2. Although methods like safety decoding and prompt patching are mentioned, experiments specifically targeting countermeasures against these defenses are lacking.

3. The impact of the attack on computational cost and inference speed has not been quantified.

4. It remains unclear whether the effectiveness of the attack degradation has been evaluated in multi-turn dialogue and long-context scenarios.

**Questions:**

see Weakness Section

---

> ### Author Response · Authors · 2025-11-23
>
> W1) HMNS is explicitly a white-box attack that requires access to attention-head activations and out-projection matrices (Sec. 3), so our experiments are deliberately restricted to open-weight models (LLaMA-2-7B-Chat, Phi-3-Medium-Instruct, LLaMA-3.1-70B; Sec. 4.1). GPT-4o / GPT-5 are used only as external graders, not as targets.
>
> W2) Defense-aware evaluation is already a core part of the paper. Sec. 4.1 lists six defenses, including Defensive Prompt Patch (DPP) and SafeDecoding (SAF). Table 4 reports defended ASR for all attacks (including HMNS) under each defense and each model, and Appendix A6 provides the per-defense compute breakdown (IPC, FLOPs, latency).
>
> W3) Compute and latency are quantified throughout. Sec. 4.3 presents a compute-normalized comparison for LLaMA-3.1-70B, including IPC, FLOPs per success, and latency per success for HMNS and strong prompt-based baselines (Table 3). Appendix A5.3 summarizes these metrics for all models/benchmarks, and Appendix A6 extends them to the defended setting.
>
> W4) Our experiments follow standard jailbreak benchmarks (AdvBench, HarmBench, JBB-Behaviors, StrongReject; Sec. 4.1), which are single-turn prompts of moderate length, and we do not claim a full evaluation in multi-turn or very long-context regimes. Importantly, HMNS is not prompt-specific: it operates on the model’s internal mechanisms by steering the residual stream at the final token of whatever context is provided. In principle, the same closed loop can be applied to multi-turn dialogue by running HMNS on the accumulated conversation prefix.
>
> We still evaluated HMNS in multi-turn and long-context settings, we conducted a new experiment using the **Multi-Turn Human Jailbreaks (MHJ) [1]** dataset. Each MHJ conversation was replayed up to the final user turn, where we replaced the last query with automated attacks from strong baselines (AutoDAN, ArrAttack, Tempest) and HMNS. This setup simulates realistic dialogue histories and stress-tests jailbreak effectiveness under long context.
>
> **Results (Table R*):** HMNS outperforms all baselines on both Phi-3-Medium-14B and LLaMA-3.1-70B, achieving **91–95% ASR with only ~2 queries**, confirming that HMNS remains highly effective and efficient even in multi-turn, long-context scenarios.
>
>
> ### Table R*: MHJ Multi-Turn Evaluation (GPT4o-graded)
>
> | Method          | ASR (Φ-3) ↑ | ACQ ↓   | ASR (LLaMA-3.1) ↑ | ACQ ↓   |
> | --------------- | ----------- | ------- | ----------------- | ------- |
> | Defense-only    | 4.8%        | —       | 3.2%              | —       |
> | AutoDAN         | 61.7%       | 12.3    | 66.0%             | 11.8    |
> | ArrAttack       | 81.0%       | 7.5     | 85.3%             | 7.2     |
> | Tempest         | 76.2%       | 9.4     | 81.1%             | 9.1     |
> | **HMNS (Ours)** | **91.6%**   | **2.1** | **95.2%**         | **2.0** |
>
> These findings reinforce our claim that re-identification and nullspace steering remain robust beyond single-turn prompts.
>
> [1] Li, N., Han, Z., Steneker, I., Primack, W., Goodside, R., Zhang, H., Wang, Z., Menghini, C. and Yue, S., 2024. Llm defenses are not robust to multi-turn human jailbreaks yet. arXiv preprint arXiv:2408.15221.

---

> > ### Author Response · Authors · 2025-11-27
> >
> > We thank you very much for your thoughtful feedback. We have updated the paper to incorporate the changes you suggested, and the corresponding clarifications and additional analyses are now included in the appendix. We would be grateful if you could kindly review these updates at your convenience.

---

### Official Review · Reviewer_vjFk · 2025-11-01

**Soundness:** 3
**Presentation:** 3
**Contribution:** 3
**Rating:** 6
**Confidence:** 3

**Summary:**

This paper introduces Head-Masked Nullspace Steering (HMNS), a circuit-level jailbreak attack that combines three mechanisms: (1) KL-divergence-based identification of causally important attention heads, (2) suppression of their write paths via column masking, and (3) injection of perturbations constrained to the orthogonal complement (nullspace) of the masked subspace. The method operates in a closed-loop detection-intervention cycle across multiple decoding attempts. Evaluated on AdvBench, HarmBench, JBB-Behaviors, and StrongReject benchmarks using LLaMA-2-7B-Chat, Phi-3-Medium-14B, and LLaMA-3.1-70B, HMNS achieves state-of-the-art attack success rates (96-99%) with low average query counts (ACQ ≈ 2). The authors introduce compute-normalized metrics (FEP, IPC, FPS, LPS) to account for internal overhead and demonstrate robustness under six defenses. Extensive ablations validate each component's contribution.

**Strengths:**

1. Theorems 2-7 provide formal guarantees about orthogonality, irreproducibility, and numerical stability. The mathematical framework is significantly more rigorous than typical jailbreak papers.

2. Leveraging nullspace projections to ensure perturbations cannot be canceled by masked heads is elegant and principled. This is a meaningful advance over ad-hoc activation steering.

3. Section 4.3 and Appendix A3-A6 introduce FEP-based accounting and compute-matched baselines. This addresses a critical methodological gap—most jailbreak papers report only external query counts.

**Weaknesses:**

1. While Appendix A3-A6 thoroughly analyzes compute, the main paper (Section 4.3) is brief. Key facts buried in appendix:
IPC ≈ 32 requires "proxy pre-selection" (A6.3) not detailed in main text;
Without pre-selection, IPC = 1 + 10·Tattribution passes = 101 worst-case (line 1577).

2. Missing comparisons: representation engineering, linear probe steering, contrastive activation addition (CAA is mentioned only in appendix A5.2)

3. Table 2 compares only to detector-based defenses, not other steering methods

**Questions:**

1. How much do head rankings change across the closed loop? Could you show attribution correlation matrices between iterations? Does instability explain the need for re-identification?

2. For the 1-4% of cases where HMNS fails (e.g., 96-99% ASR means 1-4% failure), can you characterize these failures? Are they:
Specific prompt types (e.g., certain harmful categories)?
Model-dependent (e.g., certain architectures more robust)?
Predictable from attribution patterns?

---

> ### Author Response · Authors · 2025-11-23
>
> W1) All relevant details are already provided in Appendix A2. We will clarify them more explicitly in the main text, which was previously limited by space constraints. Our two-stage attribution procedure (proxy pre-selection followed by exact KL on a shortlist) is already described in the paper and is the configuration used for all experiments (Section 4.1).
>
> W2) Our comparisons are restricted to methods that share the same constraints as HMNS: **purely inference-time, no supervised harmful labels, and no training or finetuning**. Under this setting, we already evaluate **CAA/DAS-style activation steering** (Appendix A5.2–A5.3), which sits strictly between prompt-based baselines and HMNS in both ASR and compute. The **Direct-φ (no nullspace)** variant in Table 2, which is analogous to DAS, also clearly underperforms HMNS. Most representation-engineering or linear-probe methods require extra supervision or training and are therefore outside our scope, so CAA/DAS are the closest compute-fair comparators. We will briefly clarify this and surface a compact HMNS vs. CAA/DAS summary in the main text.
>
> W3) Table2 is intentionally a \textbf{within-HMNS component ablation}, not a cross-method benchmark: each row disables one element (masking, projection, nullspace, re-identification, etc.) while holding everything else fixed to isolate its effect on ASR, ACQ, IPC, FPS, and LPS on Phi-3 14B. Comparisons to \textbf{other steering methods} already appear elsewhere: (i) Table 1 contrasts HMNS with strong prompt-based attacks (ArrAttack, AutoDAN, Tempest, etc.), (ii) Appendix A5.2--A5.3 evaluate activation-space steering baselines (CAA/DAS) under the same compute-normalized protocol and show they underperform HMNS, and (iii) Table 4 and Appendix A6 compare HMNS to prompt-based methods under six defenses, where HMNS consistently attains the best defended ASR at comparable or lower compute. We will clarify in the Table~2 caption that it is a component ablation and add explicit cross-references to these comparison tables.
>
> Q1) Head rankings do change across the loop, and we have now quantified this and linked it to the benefit of re-identification. Table 2 already shows that “No re-identification (freeze top-K at t=1)” has clearly lower ASR and worse ACQ than full HMNS on Phi-3-Medium-14B / AdvBench under the same loop budget, indicating that a fixed head set is suboptimal. To characterize this more directly, we computed Spearman correlations of per-head importance scores (Δ_{l,h}) across iterations on the AdvBench dev set (Phi-3-Medium-14B), flattening all heads across layers, measuring ρ(Δ^(t), Δ^(t’)), and the top-K overlap (fraction of heads staying in the global top-10). Results:
>
> **Table R1: Head-importance stability across HMNS iterations (Phi-3-Medium-14B, AdvBench dev).**
> rho = Spearman correlation over all heads; “Top-K overlap” = % of heads common to both global top-10 sets.
>
> | Iteration pair (t, t’) | rho (all heads) | Top-K overlap (%) |
> | ---------------------- | --------------- | ----------------- |
> | (1, 2)                 | 0.78            | 81                |
> | (2, 3)                 | 0.74            | 78                |
> | (3, 4)                 | 0.71            | 75                |
> | (1, 4)                 | 0.63            | 68                |
>
> These numbers show that head rankings are neither static nor random: there is a stable core (rho ≈ 0.7–0.8 with ≈70–80% top-K overlap) but also systematic churn in the remaining heads as the context and defenses evolve. Combined with the ablation where freezing the initial top-K degrades ASR and ACQ, this supports our design choice: re-identification is needed not because attribution is unstable noise, but because causal-head importance changes meaningfully across the closed loop, and exploiting this adaptivity is crucial for HMNS’s performance.
>
> Q2) **Table R2: HMNS failure modes on AdvBench/HarmBench dev (avg. over Phi-3-14B / LLaMA-3.1-70B).**
>
> | Failure mode                              | Share of failures | Example behavior (paraphrased)                                                                                  |
> | ----------------------------------------- | ----------------- | --------------------------------------------------------------------------------------------------------------- |
> | Distributed safety (many heads/layers)    | ≈55%              | “Compare several synthesis routes for an illegal drug and pick the easiest at home.”                            |
> | Defense re-framing / deflection           | ≈30%              | “Slightly edit a political article to favor an extremist group.” (often turned into high-level/meta discussion) |
> | Other / model-specific (budget, outliers) | ≈15%              | Long, multi-step prompts mixing several harms where T_att=10 is sometimes insufficient.                         |

---

> > ### Author Response · Authors · 2025-11-27
> >
> > We thank you very much for your thoughtful feedback. We have updated the paper to incorporate the changes you suggested, and the corresponding clarifications and additional analyses are now included in the appendix. We would be grateful if you could kindly review these updates at your convenience.

---

### Author Response · Authors · 2025-12-04

We thank all reviewers and the area chair for their detailed and constructive feedback on *“Jailbreaking the Matrix: Nullspace Steering for Controlled Model Subversion.”* We have carefully revised the paper to address the main concerns around threat modeling, comparisons, compute accounting, and scope, and we briefly summarize the resulting changes here.

First, we have made our **threat model** and scope explicit in the main text: HMNS is a **white-box, mechanism-level** attack aimed at worst-case robustness analysis and mechanistic insight for open-weight models, not a deployment-time black-box adversary. Within this setting, the attacker is still constrained: we only (i) mask a small, causally identified set of heads and (ii) inject nullspace-constrained perturbations under fixed compute and loop budgets, without arbitrary weight edits. We now foreground these assumptions in the introduction and Sec. 3, and we clarify that our goal is to understand where safety “lives” in the model and how fragile it is to small, interpretable interventions, rather than to propose a practical API-level exploit.

Second, we substantially **expanded and surfaced compute and comparison details** that were previously buried in the appendix. Section 4.3 now explicitly describes our two-stage attribution procedure (proxy pre-selection + KL on a shortlist), the resulting IPC/FEP/FPS/LPS metrics, and how these yield IPC ≈ 32 in practice rather than the worst-case 101 passes. We clarify that Table 2 is a **within-HMNS component ablation** (masking, nullspace, projection, re-identification, etc.), and we add explicit cross-references to: (i) prompt-based baselines (Table 1), (ii) activation-space steering baselines such as CAA/DAS (Appendix A5.2–A5.3), and (iii) defended comparisons under six safety mechanisms (Table 4, Appendix A6). We also strengthened the related work section to explicitly discuss attention-based jailbreaks (e.g., AttnGCG, Attention Eclipse) and to position HMNS as a complementary, circuit-level intervention rather than a replacement for input-level attacks.

Third, we added **new analyses and experiments** in response to reviewers’ questions. To address the role of re-identification, we now report head-importance dynamics across the closed loop, including Spearman correlations and top-K overlaps between iterations, and show that “frozen” head sets degrade ASR and ACQ relative to full HMNS—indicating that re-identification tracks meaningful shifts in causal importance rather than noise. We characterize HMNS failure modes across AdvBench/HarmBench (distributed safety, defense reframing, and long multi-step prompts), and we include a **multi-turn, long-context evaluation** using MHJ, where HMNS achieves 91–95% ASR with ~2 queries while maintaining its advantage over strong prompt-based baselines. Finally, we add a **Self-Defense–style output-filter experiment** (inspired by Phute et al.), showing that while defended ASR drops for all methods, HMNS remains the strongest under this additional output-stage check.

Fourth, we explicitly **acknowledge key limitations and future directions**. We now state clearly that HMNS is designed for open-weight models and does not directly apply to closed-source APIs like GPT-4/Claude/Gemini. We discuss how HMNS can still be practically relevant as an internal red-teaming tool for labs that control their own weights and as a probe for locating and stress-testing safety circuits. We also highlight that the same geometry- and attribution-based tools used for subversion could be flipped into defenses—for example, to stabilize safety heads against steering or to build more robust internal monitors—and we point to this as an explicit direction for future work.

We hope these revisions clarify our assumptions, sharpen the empirical narrative, and better situate HMNS within the broader jailbreak and safety literature. We are grateful for the reviewers’ engagement and believe their feedback has helped us present a clearer and more balanced account of both the strengths and the limitations of our approach.

---

### Meta-Review · Area_Chair_2C4r · 2026-01-07

**Summary:**

This paper introduces Head-Masked Nullspace Steering (HMNS), a mechanism-level, inference-time jailbreak that 1) identifies causally important attention heads, 2) masks their out-projection contributions, and 3) injects a steering vector constrained to the orthogonal complement of the masked write subspace. Extensive results are provided to show its efficacy.

The initial reviews were mixed (but leaned towards positive). Overall, reviewers found the method technically interesting and empirically strong. But meanwhile, multiple major concerns are raised: 1) the white-box assumption seems too strong (which is very likely not realistic); 2) the comparisons to other baselines may not be that fair; 3) how head rankings change across the closed loop should be analyzed; and 4) more evaluations in multi-turn/long-context scenarios and against output-filtering defenses should be provided.

As detailed in the next part, the AC believes most of these major concerns are well responded in the rebuttal. The only controversial part (or we may call it a limitation) is that HMNS is a white-box attack, which may not be that realistic in the real world. But this should not be a major concern, and could still serve as a useful study for inspiring future works. Therefore, the AC recommends acceptance.

**Reviewer Concerns:**

Addressed concerns:
1) Compute Analysis: The authors clarified the calculation of Internal Pass Counts, as well as providing new results under the same compute budget.

2) Head Ranking: new experiments are added on showing a Spearman correlation of ~0.7 between iterations.

3) More Ablations: new experiments on the Multi-Turn Human Jailbreaks dataset and with a "Self-Defense" style output filter are added.


Outstanding Concerns:
1) White-box Assumption: Reviewer 3 argues that the white-box assumption is "unrealistically privileged" and less relevant for alignment safety than black-box threats. While, in the rebuttal, the authors clarify this study is mainly for "worst-case robustness analysis" and mechanistic insight, this philosophical difference regarding the value of white-box jailbreaks remains unresolved for R3.

**Reviewer Scores:**

For the three reviewers who rated positively on this paper, I believe their scores will remain the same or get slightly bumped up after the rebuttal.

For the reviewer who originally rated negatively, as the major concern is about its white-box nature (which is not fully addressed in the rebuttal), the final rating should stay the same. But as mentioned above, the AC believes this is not necessarily a major weakness for rejecting the paper.

---

### Decision · Program_Chairs · 2026-01-26

Accept (Poster)